# Optimizing Inference-Time Compute for Medical Reasoning via Uncertainty Quantification

Shaohao Rui [1 2]   Kaitao Chen [3 4]   Weijie Ma [2 3]   Xiaosong Wang [2 4]

## Abstract

Extended Chain-of-Thought (CoT) reasoning has significantly bolstered the capabilities of medical large language models (LLMs). However, current models exhibit static computational expenditure, applying lengthy reasoning processes indiscriminately to both simple queries and complex diagnostic cases. This inefficiency is particularly prohibitive in real-world healthcare, where clinical scenarios range from time-sensitive emergencies requiring rapid response to intricate pathologies demanding deep analysis. To address this, we propose **AdaThink-Med**[1], an end-to-end framework for adaptive reasoning via uncertainty-guided length calibration. Although the underlying mechanism is generalizable, we demonstrate its critical value in the medical domain, where balancing inference latency with diagnostic precision is paramount. AdaThink-Med leverages entropy-based uncertainty estimation within reinforcement fine-tuning to dynamically shape reward signals: it penalizes verbosity for high-confidence correct answers (e.g., straightforward knowledge retrieval) while incentivizing extended exploration for uncertain or ambiguous scenarios. Across six medical benchmarks, AdaThink-Med reduces inference token consumption by **4.7× to 6.4×** on Qwen and Llama architectures, respectively, with minimal performance trade-offs. Notably, our reward design naturally produces distinct "non-thinking" and "thinking" modes within a single model, enabling efficient allocation of computational resources without any external router or classifier.

## 1. Introduction

Recent advances in Chain-of-Thought (CoT) reasoning (Wei et al., 2022) and reinforcement learning (Suzgun et al., 2022; Zhou et al., 2022; Chen et al., 2024a; Guo et al., 2025) have significantly improved the reasoning abilities of LLMs with leveraging extended trial-and-error processes and reflective thinking—also known as inference time scaling. Such progress has inspired new paradigms for building general-purpose medical AI systems. Through high-quality long CoT data for supervised fine-tuning or rule-based reinforcement learning, substantial gains have been made in complex clinical tasks including differential diagnosis, surgical planning, and prognosis prediction (Chen et al., 2024b; Huang et al., 2025; Liu et al., 2025a; Rui et al., 2026; 2025; Chen et al., 2025b).

However, such improvements often come at the cost of significantly increased lengthy outputs, thereby raising inference time and costs. For example, when answering a simple question like "1 + 1 = ?", models such as DeepSeek-R1 (Guo et al., 2025) have been observed to produce outputs as long as 170 tokens. In parallel, another line of research has identified the *overthinking* phenomenon (Chen et al., 2024c; Luo et al., 2025), where reasoning models generate unnecessarily long chains of thought, which can even degrade performance on simple questions (Su et al., 2025). Moreover, (Zhang et al., 2025) reports that up to 58% of questions in existing medical benchmarks can be accurately answered with short, concise outputs—without resorting to expensive extended reasoning. These findings collectively underscore the urgent need to improve the efficiency of reasoning models, especially if they are to be deployed effectively in real-world applications.

Most existing work aims to improve the efficiency of general reasoning models by applying length-based reward, which penalizes outputs with long CoT. Specifically, (Team et al., 2025; Yu et al., 2025) encourages shorter outputs across the board, while another line of work (Shen et al., 2025; Yi et al., 2025; Yeo et al., 2025) penalizes long outputs when the answer is correct but encourages longer outputs for incorrect ones to allow more exploration. However, these methods rely solely on correctness and largely ignore the underlying problem difficulty, which limits their ability to achieve

---

[1]Shanghai Jiao Tong University, Shanghai, China [2]Shanghai Innovation Institute, Shanghai, China [3]Fudan University, Shanghai, China [4]Shanghai AI Laboratory, Shanghai, China. Correspondence to: Xiaosong Wang <wangxiaosong@pjlab.org.cn>.

[1]Code available at https://github.com/shaohao011/AdaThinkMed

optimal efficiency improvements. Ideally, from a human perspective, computational resources should be allocated adaptively, i.e., spending more effort on complex questions and less on simpler ones. Recent studies further suggest that difficult questions often benefit from longer CoT to maintain high performance, whereas simple problems can be solved with much shorter outputs or even without explicit reasoning (Liu et al., 2025b; Ma et al., 2025; Jiang et al., 2025; Fang et al., 2025). In the medical domain, pioneering work (Zhang et al., 2025) decomposes the reasoning process into thinking and non-thinking modes based on the estimated difficulty of each question. However, it relies on manual annotations to train a difficulty classifier, introducing scalability bottlenecks and potential labeling bias.

Furthermore, recent studies highlight a persistent trade-off between performance and inference cost in hybrid adaptive models. For instance, Qwen3 exhibits notable performance degradation in its hybrid architecture particularly for non-thinking tasks—compared to specialized dual-model setups, prompting a strategic reversion to separate models for reasoning-intensive versus routine queries (Team, 2025; The Register, 2025). Similarly, while DeepSeekV3.1 (DeepSeek-AI, 2024) and GPT-5 adopt hybrid architectures, they necessitate vast additional data to mitigate performance loss, yet user feedback still favors the specialized experience of dual models. Consequently, achieving an optimal balance within a single unified model remains challenging, underscoring the urgent need for more effective optimization strategies.

Motivated by the above observations, we propose AdaThink-Med, the first end-to-end framework designed to improve adaptive thinking capabilities in medical language models by explicitly incorporating problem difficulty. The central idea of AdaThink-Med is to enable the model to generate outputs of varying lengths depending on the difficulty of the input question, producing concise answers for simple cases and long reasoning chains for complex ones. During training, for each question, we generate multiple outputs and evaluate both their accuracy and uncertainty to derive a comprehensive measure of difficulty. This difficulty-aware signal is then incorporated into the reinforcement learning process, guiding the model to generate shorter outputs for low-difficulty with high-confidence right answer. At inference time, AdaThink-Med automatically estimates the difficulty of incoming questions and generates outputs with an appropriate amount of reasoning.

We conduct comprehensive experiments across both rule-based and model-based RL paradigms, and conduct the evaluation on six public medical reasoning benchmarks spanning various difficulty levels. Results show that AdaThink-Med consistently achieves better efficiency compared to existing medical language models and length-based output control methods.

The contributions of this work are fourfold:

- We propose **AdaThink-Med**, an end-to-end framework that allows a single medical LLM to switch between reasoning modes adaptively based on problem difficulty, without any external router or classifier.

- We introduce an entropy-guided length calibration mechanism whose reward design naturally produces distinct "non-thinking" (direct response) and "thinking" (extended reasoning) modes within one model.

- We demonstrate the framework's utility in dataset selection, facilitating the efficient extraction of high-quality training subsets from large-scale corpora.

- We achieve state-of-the-art performance across six medical benchmarks, delivering a superior trade-off between diagnostic accuracy and computational cost.

## 2. Related Works

**Efficient Reasoning via Reinforcement Learning.** Efficient reasoning seeks to mitigate the "overthinking phenomenon" by optimizing the trade-off between accuracy and computational cost (Sui et al., 2025). Prior works, such as Kimi1.5 (Team et al., 2025), DAPO (Yu et al., 2025), and ShortBetter (Yi et al., 2025), primarily rely on greedy length penalties or minimal-length targeting. However, these strategies often lead to reward hacking and performance degradation, as they fail to incentivize extended reasoning for complex problems. While approaches like CosFn (Yeo et al., 2025) and DAST (Shen et al., 2025) attempt to address this by incorporating correctness or static difficulty, they generally overlook the inherent uncertainty in reasoning or rely on inflexible offline training. **In contrast**, our method introduces an entropy-based uncertainty measure for dynamic difficulty estimation within an online learning framework, enabling adaptive thinking that continuously evolves with the model's capabilities.

**Entropy-Guided Reasoning in LLMs.** Recent research has increasingly leveraged entropy as a proxy for uncertainty to steer reasoning processes. Notable approaches include using semantic entropy for sample weighting in RL (Chen et al., 2025c), dynamic inference termination (Xu et al., 2025), or reasoning trace compression (Li et al., 2025; Zhu et al., 2025). Distinct from these works which primarily focus on re-weighting or explicit pruning, **AdaThink-Med** proposes a specific entropy-based difficulty estimator that fuses correctness with uncertainty. We integrate this estimator into a difficulty-aware length reward mechanism during reinforcement fine-tuning. This design dynamically penalizes redundancy for easy queries while incentivizing exploration for hard ones, rather than simply shortening outputs. To our knowledge, this represents the first application of such an

entropy-guided mechanism to medical RL, where our reward design naturally produces reasoning mode separation and additionally enables effective dataset selection.

## 3. Method

### 3.1. Overview of AdaThink-Med

The core objective of AdaThink-Med is to transform a static medical reasoning model into an adaptive model that dynamically allocates computational resources based on clinical complexity. Formally, we aim to maximize the expected reward while minimizing the inference cost, regularized by a reference policy (see Appendix C).

Unlike prior approaches that impose uniform length penalties, AdaThink-Med introduces an uncertainty-guided length calibration mechanism. As illustrated in Figure 1, the framework operates in two phases. First, for a given medical query, the model generates multiple reasoning trajectories. We estimate the problem difficulty by jointly analyzing the correctness of these responses and their intrinsic predictive uncertainty, quantified via entropy. Second, during Adaptive Reinforcement Fine-Tuning, this difficulty signal modulates the reward function. For simple queries characterized by high confidence and correctness, the model is penalized for redundancy. Conversely, for complex queries exhibiting high uncertainty or errors, the model is incentivized to extend its reasoning chain to explore alternative solutions.

### 3.2. Uncertainty-Guided Difficulty Estimation

A robust adaptive reasoning model must accurately distinguish between simple and complex queries. We propose that incorporating output-level uncertainty into the difficulty assessment is essential, as reliance solely on sampling accuracy overlooks the inherent variability of LLM generation.

**Output Uncertainty Calculation.** Following prior work (Wang et al., 2025), we quantify token-level uncertainty using Shannon entropy. For a generated sequence, the entropy $H_t$ at decoding step $t$ is defined as:

$$H_t := -\sum_{j=1}^{V} p_{t,j} \log p_{t,j},$$

$$\text{s.t. } p_t = \pi_\theta(\cdot \mid \mathbf{x}, s_{<t}) = \text{Softmax}\left(\frac{z_t}{T}\right), \quad (1)$$

where $V$ is the vocabulary size, $z_t$ represents the logits, and $T$ is the temperature. We evaluate entropy using the training policy $\pi_\theta$ distribution. To represent the uncertainty of the entire trajectory $o_i$, we calculate the mean entropy of the top-$K$ most uncertain tokens, denoted as $\mathcal{H}_i$. Guided by (Wang et al., 2025), which suggests that the top 20% of high-entropy tokens effectively represent sequence-level uncertainty, we adopt this ratio as our default setting for $K$.

We further investigate the sensitivity of $K$ through ablation studies in Appendix 9. To ensure numerical stability across samples within a batch $\mathcal{B}$, we apply min-max normalization to these trajectory-level uncertainties:

$$\mathcal{H}_i := \frac{1}{K} \sum_{t \in \mathcal{T}_K} H_t,$$

$$\widetilde{\mathcal{H}}_i := \frac{\mathcal{H}_i - \min_j \mathcal{H}_j}{\max_j \mathcal{H}_j - \min_j \mathcal{H}_j}, \quad (2)$$

$$\forall i \in \{1, \ldots, |\mathcal{B}| \times G\},$$

where $\mathcal{T}_K$ is the set of indices corresponding to the top-$K$ entropy values in the sequence.

**Difficulty Estimation.** We define problem difficulty by integrating correctness with the normalized uncertainty. For a question with ground-truth $y^*$ and $G$ sampled rollouts, the difficulty score $\mathcal{D}_q$ is formulated as:

$$\mathcal{D}_q = 1 - \frac{1}{G} \sum_{i=1}^{G} \mathbb{I}[o_i = y^*] \cdot \Big(\alpha(1 - \widetilde{\mathcal{H}}_i) + (1 - \alpha)\Big). \quad (3)$$

Here, $\mathbb{I}[\cdot]$ is the indicator function and $\alpha$ balances the contribution of uncertainty. This formulation assigns higher difficulty scores to questions where the model is incorrect, or where it yields correct answers with high uncertainty.

### 3.3. Difficulty-Aware Length Calibration

To enforce adaptive reasoning, we modulate the reward based on the estimated difficulty relative to a dynamic batch threshold.

**Dynamic Thresholding.** We compute a difficulty threshold $\theta_\mathcal{B}$ as the $\tau$-quantile of difficulty scores within the batch $\mathcal{B}$. To ensure training stability, we apply an exponential moving average (EMA) update to the threshold and the average lengths for simple ($\bar{L}_s$) and hard ($\bar{L}_h$) questions:

$$\theta_\mathcal{B} \leftarrow \gamma \cdot \theta_\mathcal{B}^{\text{cur}} + (1 - \gamma) \cdot \theta_\mathcal{B}^{\text{old}}, \quad (4)$$

where $\gamma$ is the momentum coefficient.

**Length Reward.** The length-based reward $\mathcal{R}_{\text{len}}^i$ is defined piecewise to penalize redundancy in easy tasks and encourage depth in hard tasks:

$$\mathcal{R}_{\text{len}}^i = \begin{cases} \max\left(0, 1 - \frac{L_i}{\bar{L}_s} \cdot \rho_i\right), & \text{if } \mathcal{D}_q < \theta_\mathcal{B} \wedge o_i = y^*, \\ \min\left(1, \frac{L_i}{\bar{L}_h} \cdot \rho_i - 1\right), & \text{if } \mathcal{D}_q > \theta_\mathcal{B} \wedge o_i \neq y^*, \\ 0, & \text{otherwise.} \end{cases} \quad (5)$$

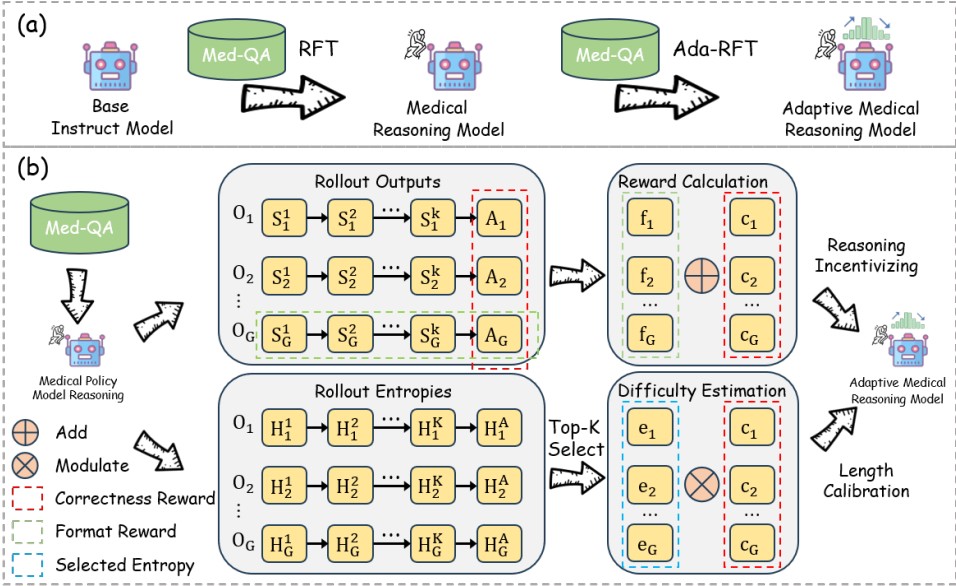

*Figure 1.* **Overview of the AdaThink-Med framework.** (a) **Staged Adaptive Training:** The training process consists of two stages. First, a Base Instruct Model is trained via standard Reinforcement Fine-Tuning (RFT) to establish medical reasoning capabilities. Subsequently, the model undergoes Adaptive RFT (Ada-RFT) to optimize reasoning efficiency. (b) **Uncertainty-Guided Difficulty Estimation and Length Calibration:** During the adaptive phase, the model generates multiple rollout outputs ($O_1 \ldots O_G$). The framework simultaneously calculates standard rewards (correctness and format) and estimates output uncertainty via **Top-K token entropies**. By combining correctness ($c_i$) and uncertainty ($e_i$), the system estimates problem difficulty, which dynamically calibrates the length reward to incentivize concise answers for simple questions and extended reasoning for complex ones.

where $L_i$ is the length of output $o_i$, and $\rho_i = 0.5 + 0.5(1 - \widetilde{\mathcal{H}}_i)$ is a scaling factor dependent on uncertainty. The first condition discourages unnecessarily long outputs for simple, correctly answered questions. The second condition provides performance compensation by rewarding longer reasoning chains for difficult questions where the model initially fails. The remaining two quadrants—*Hard+Correct* and *Easy+Incorrect*—are deliberately assigned a zero length signal so that brevity is never penalized when a fragile reasoning chain happens to succeed, and length is never rewarded when the model has not yet produced a correct answer; in both regimes the accuracy reward $\mathcal{R}_{\text{acc}}$ dominates the gradient. Empirically, this asymmetric design is what prevents the collapse modes observed in greedy length-calibration baselines (Section 4.1); a fuller ablation is provided in Appendix F.1.

The total reward for each trajectory combines the accuracy reward $\mathcal{R}_{\text{acc}}$, format reward $\mathcal{R}_{\text{format}}$, and the adaptive length reward:

$$\mathcal{R}_i = \mathcal{R}_{\text{acc}}^i + \lambda_1 \mathcal{R}_{\text{format}}^i + \lambda_2 \mathcal{R}_{\text{len}}^i. \tag{6}$$

We employ default values $\lambda_1 = \lambda_2 = 0.5$. This composite reward is then utilized to optimize the policy via Group Relative Policy Optimization (GRPO).

## 4. Experiments

**Training Datasets and Evaluation Benchmarks.** We use the AlphaMed19k (Liu et al., 2025a) dataset as our primary training corpus. AlphaMed19k integrates the official training splits (Huang et al., 2025) of two public multiple-choice medical QA benchmarks, MedQA (Jin et al., 2021) and MedMCQA (Pal et al., 2022). MedQA contains expert-level clinical questions drawn from the USMLE, while MedM-CQA covers factoid and reasoning questions from Indian medical entrance examinations. For out-of-domain evaluation we use PubMedQA (Jin et al., 2019), the medical subsets of MMLU-Pro (Wang et al., 2024) (MMLU-ProM), the medical subsets of GPQA (Rein et al., 2023) (GPQA-M), and the recent clinical benchmark MedXpertQA (Zuo et al., 2025). Additional dataset details are provided in the Appendix E.3.

**Baseline & Implementation Details.** We compare AdaThink-Med with baseline models from three categories: general zero-shot LLMs (Qwen-2.5-Instruct-7B, Llama-3.1-Instruct-8B), medical tuned LLMs (MedLlama3, MMed, Med42 (Christophe et al., 2024), OpenBioLLM (Ankit Pal, 2024), UltraMedical (Zhang et al., 2024), HuatuoGPT-o1 (Chen et al., 2024b), m1 (Huang et al., 2025)), and length-calibration methods (Kimi1.5 (Team et al., 2025), ShortBetter (Yi et al., 2025), DAST (Shen et al., 2025), CosFn (Yeo et al., 2025)). Because the $G$ rollouts required by our diffi-

culty estimator are already produced by vanilla GRPO and entropy is computed from existing logits, AdaThink-Med adds less than 1% wall-clock overhead per training step. More details of baselines and implementation are summarized in Appendix E.1 and Appendix E.2, respectively.

**Evaluation Metrics.** We report accuracy and output length to assess correctness against ground truth and token usage, respectively. To capture the trade-off between computational cost and performance, we adopt the **Accuracy-Efficiency Score (AES)** (Yi et al., 2025; Luo et al., 2025), defined as:

$$\text{AES} = \begin{cases} \alpha \cdot \Delta\text{Length} + \beta \cdot |\Delta\text{Acc}|, & \text{if } \Delta\text{Acc} \geq 0, \\ \alpha \cdot \Delta\text{Length} - \gamma \cdot |\Delta\text{Acc}|, & \text{if } \Delta\text{Acc} < 0. \end{cases}$$

Here, $\Delta\text{Length} = \frac{\text{Length}_{\text{baseline}} - \text{Length}_{\text{model}}}{\text{Length}_{\text{baseline}}}$ is the relative reduction in output length and $\Delta\text{Acc} = \frac{\text{Acc}_{\text{model}} - \text{Acc}_{\text{baseline}}}{\text{Acc}_{\text{baseline}}}$ is the relative change in accuracy. **We set** $\alpha = 1$, $\beta = 3$, **and** $\gamma = 5$ **by default for fair comparison.** The asymmetric design places a larger penalty on accuracy drops than the reward for improvements, which aligns with practical preferences.

## 4.1. Results

**Performance Comparison with General and Medical Reasoning LLMs.** We evaluate AdaThink-Med against leading general and medical reasoning models to assess inference efficiency. As detailed in Table 1, our approach achieves substantial efficiency gains across all six benchmarks. Most notably, AdaThink-Med realizes a 4.7× compression for the Qwen backbone (reducing average length from 497 to 106 tokens) with a slight accuracy improvement (+0.25%), and a 6.4× compression for the Llama backbone (410 to 64 tokens) with minimal performance degradation (-1.13%). Unlike baselines that rely on extensive datasets and heavy inference costs, AdaThink-Med establishes state-of-the-art efficiency using only 19k samples without distilled CoT data. Furthermore, the model exhibits clear adaptive behavior: it generates significantly shorter responses for straightforward tasks (e.g., MedQA) while allocating more computational resources to complex reasoning challenges (e.g., GPQA-M), validating the proposed adaptive thinking strategy.

**Length Reward Hack & Motivation of AdaThink-Med's Performance Compensation.** We conduct pilot experiments to investigate greedy output length reduction strategies that do not explicitly incentivize incorrect responses. Two representative approaches are Kimi1.5 (Team et al., 2025), which enforces uniform length reduction across all samples, and ShortBetter (Yi et al., 2025), which drives responses toward the minimal length among correct samples. As shown in Fig. 2(a), the absence of such compensation causes both Kimi1.5 and ShortBetter to collapse in terms

of output length and accuracy. We initially suspect that this collapse primarily arises in close-ended tasks, where the presence of candidate answers allows the model to adopt a trivial strategy of directly outputting the final answer without any reasoning. To avoid this shortcut, we adopt an open-ended RL setting following HuatuoGPT-o1 (Chen et al., 2024b), where the model must reason and answer using its world knowledge without being provided candidate options (Rui et al., 2025). Implementation details are given in the Appendix. As shown in Fig. 2(b), even in the open-ended setting, greedy length calibration leads to collapse, with both accuracy and output length dropping sharply and failing to recover. Moreover, we found that the collapsed model no longer produces meaningful answers or follows instructions. We attribute this phenomenon to a form of length reward hack: in its search for shorter responses, the model converges to a minimal-length strategy that maintains apparent accuracy while maximizing length-based rewards. This shortcut undermines the emergence of diverse reasoning patterns and degrades instruction following. To address this issue, we introduce a performance compensation mechanism that penalizes shorter outputs when accuracy degrades. This encourages the model to achieve a more balanced trade-off between performance and efficiency, ultimately leading to stable training dynamics as shown in Fig. 2.

**Comparison with Length-calibration Methods.** As shown in Table 2, AdaThink-Med delivers marked efficiency gains over existing length-calibration baselines. It reduces the average reasoning length to 64 tokens on Llama and 106 tokens on Qwen. These lengths are 2.0× and 2.8× shorter than those produced by Kimi1.5, and 2.6× and 3.2× shorter than those from ShortBetter, respectively, while achieving comparable accuracy (Llama: 55.59% vs. 55.94% for Short-Better; Qwen: 54.99% vs. 54.28% for ShortBetter). Additionally, AdaThink-Med achieves the highest AES scores on both model backbones, with 0.92 on Llama and 0.93 on Qwen, respectively. This corresponds to an improvement of +0.14 over the next best method (Kimi1.5) on Llama and +0.38 on Qwen. These improvements stem from the model's adaptive thinking capability: AdaThink-Med dynamically allocates reasoning length, maintaining brevity on easy questions while employing extended reasoning chains on challenging or initially incorrect cases through its accuracy aware compensation mechanism. This approach effectively reduces computational cost without sacrificing reliability.

**Reasoning Mode Separation via Reward Design.** As illustrated in Fig. 3(a), models trained with AdaThink-Med exhibit a notable bimodal distribution in reasoning length, separating into two behaviors: a *non-thinking* mode that outputs answers directly without intermediate reasoning, and a *thinking* mode characterized by concise yet essential reasoning steps. We do not claim this as a spontaneously

*Table 1.* **Performance of models on in-domain (\*) and out-of-domain medical QA benchmarks.** The best results are shown in **bold**. + indicates reasoning ability obtained with a Chain-of-Thought prompt; † denotes supervised fine-tuning with long CoT data; ◇ represents reinforcement learning; and ‡ refers to training with additional datasets. Gray rows indicate baseline reasoning models prior to adaptive optimization, while yellow rows denote the final **AdaThink-Med** adaptive models. For AES, the baseline is Llama-3.1-8B-Instruct+.

| Model | MedQA* | | MedMCQA* | | PubMedQA | | MMLU-ProM | | GPQA-M | | MedXpert | | Avg | | AES |
|---|---|---|---|---|---|---|---|---|---|---|---|---|---|---|---|
| | Acc. | Len. | Acc. | Len. | Acc. | Len. | Acc. | Len. | Acc. | Len. | Acc. | Len. | Acc. | Len. | |
| *Llama backbone* | | | | | | | | | | | | | | | |
| Llama-3.1-8B-Instruct+ | 68.18 | 461 | 57.32 | 309 | 78.20 | 296 | 60.06 | 457 | 45.64 | 485 | 16.08 | 545 | 54.25 | 425 | – |
| Llama-3.1-8B-Instruct | 54.75 | 302 | 54.69 | 169 | 77.00 | 113 | 57.00 | 256 | 35.64 | 345 | 14.48 | 342 | 48.93 | 254 | -0.09 |
| MedLlama3-8B-v1+ | 50.90 | 187 | 41.47 | 153 | 46.00 | 133 | 30.68 | 234 | 32.05 | 265 | 12.20 | 310 | 35.55 | 214 | -1.23 |
| MedLlama3-8B-v2+ | 60.56 | 271 | 54.53 | 254 | 73.40 | 216 | 54.26 | 323 | 43.84 | 378 | 13.91 | 315 | 50.08 | 293 | -0.07 |
| MMed-8B†‡+ | 49.01 | 888 | 43.82 | 843 | 59.40 | 832 | 34.65 | 856 | 31.79 | 831 | 13.06 | 938 | 38.62 | 865 | -2.47 |
| Med42-8B‡+ | 57.97 | 343 | 55.53 | 276 | 70.80 | 204 | 53.28 | 357 | 43.07 | 359 | 13.26 | 402 | 48.99 | 324 | -0.25 |
| OpenBioLLM-8B†‡◇ | 50.51 | 183 | 44.84 | 160 | 55.40 | 141 | 38.82 | 153 | 31.28 | 167 | 12.16 | 207 | 38.84 | 169 | -0.82 |
| UltraMedical-8B-3†‡◇ | 70.30 | 402 | 59.16 | 364 | 76.80 | 282 | 59.93 | 524 | 46.92 | 490 | 15.91 | 630 | 54.84 | 449 | -0.02 |
| UltraMedical-8B-3.1†‡◇ | 74.46 | 486 | 62.65 | 450 | 78.50 | 392 | 62.93 | 603 | 46.92 | 588 | 16.89 | 710 | 57.06 | 538 | -0.11 |
| HuatuoGPT-o1-8B†‡◇ | **76.35** | 568 | **62.82** | 445 | **79.80** | 445 | 63.71 | 521 | **54.35** | 585 | 17.06 | 601 | **59.02** | 527 | +0.02 |
| GRPO-LLama◇ | 72.50 | 438 | 61.60 | 299 | 78.50 | 290 | 63.51 | 442 | 46.92 | 461 | 17.26 | 529 | 56.72 | 410 | +0.17 |
| **AdaThink-Med-LLama** | 67.00 | **36** | 60.41 | **73** | 78.40 | **54** | 63.58 | **85** | 47.94 | **100** | 16.20 | **35** | 55.59 | **64** | **+0.92** |
| *Qwen backbone* | | | | | | | | | | | | | | | |
| Qwen2.5-7B-Instruct+ | 63.86 | 490 | 56.68 | 359 | 73.10 | 386 | 62.28 | 523 | 46.41 | 562 | 12.28 | 585 | 52.44 | 484 | - |
| Qwen2.5-7B-Instruct | 54.28 | 313 | 53.43 | 196 | 72.70 | 129 | 56.67 | 292 | 38.71 | 448 | 12.44 | 291 | 48.04 | 278 | -3.73 |
| m1-7B†‡ | 75.01 | 2161 | 62.32 | 1749 | 74.20 | 1080 | **68.07** | 2564 | 51.53 | 3887 | **18.28** | 3265 | 58.24 | 2451 | -4.54 |
| GRPO-Qwen◇ | 67.55 | 504 | 59.52 | 378 | 72.30 | 396 | 66.12 | 535 | 48.46 | 575 | 14.48 | 592 | 54.74 | 497 | +0.11 |
| **AdaThink-Med-Qwen** | 68.34 | 79 | 58.74 | 109 | 73.50 | 62 | 66.45 | 121 | 48.72 | 175 | 14.16 | 88 | 54.99 | 106 | +0.93 |

*Table 2.* **Performance of length calibration methods on in-domain (\*) and out-of-domain medical QA benchmarks.** For AES, baseline model is LLama-3.1-8B-Instruct+ and Qwen-2.5-7B-Instruct+, respectively.

| Model | MedQA* | | MedMCQA* | | PubMedQA | | MMLU-ProM | | GPQA-M | | MedXpert | | Avg | | AES |
|---|---|---|---|---|---|---|---|---|---|---|---|---|---|---|---|
| | Acc. | Len. | Acc. | Len. | Acc. | Len. | Acc. | Len. | Acc. | Len. | Acc. | Len. | Acc. | Len. | |
| *Llama backbone* | | | | | | | | | | | | | | | |
| Llama-3.1-8B-Instruct+ | 68.18 | 461 | 57.32 | 309 | 78.20 | 296 | 60.06 | 457 | 45.64 | 485 | 16.08 | 545 | 54.25 | 425 | – |
| Llama-3.1-8B-Instruct | 54.75 | 302 | 54.69 | 169 | 77.00 | 113 | 57.00 | 256 | 35.64 | 345 | 14.48 | 342 | 48.93 | 254 | -0.09 |
| Llama-3.1-8B-Instruct-GRPO | 72.50 | 438 | 61.60 | 299 | 78.50 | 290 | 63.51 | 442 | 46.92 | 461 | **17.26** | 529 | **56.72** | 410 | +0.17 |
| Kimi1.5 (Team et al., 2025) | 70.62 | 145 | 60.93 | 100 | 77.50 | 92 | **63.71** | 164 | 45.38 | 118 | 15.75 | 131 | 55.65 | 125 | +0.78 |
| CosFn (Yeo et al., 2025) | 37.94 | 1007 | 49.29 | 919 | 75.10 | 885 | 23.32 | 998 | 32.82 | 1012 | 10.12 | 1020 | 38.10 | 974 | -2.78 |
| DAST (Shen et al., 2025) | 60.64 | **30** | 56.9 | **34** | 76.00 | **39** | 59.48 | **71** | 47.69 | **53** | 15.35 | **82** | 52.71 | **51** | +0.74 |
| ShortBetter (Yi et al., 2025) | 71.48 | 175 | 60.62 | 134 | **78.60** | 118 | 62.99 | 188 | 45.89 | 219 | 16.08 | 171 | 55.94 | 167 | +0.70 |
| **AdaThink-Med-LLama** | 67.00 | 36 | 60.41 | 73 | 78.40 | 54 | 63.58 | 85 | **47.94** | 100 | 16.20 | 35 | 55.59 | 64 | **+0.92** |
| *Qwen backbone* | | | | | | | | | | | | | | | |
| Qwen2.5-7B-Instruct+ | 63.86 | 490 | 56.68 | 359 | 73.10 | 386 | 62.28 | 523 | 46.41 | 562 | 12.28 | 585 | 52.44 | 484 | – |
| Qwen2.5-7B-Instruct | 54.28 | 313 | 53.43 | 196 | 72.70 | 129 | 56.67 | 292 | 38.71 | 448 | 12.44 | 291 | 48.04 | 278 | +0.01 |
| Qwen2.5-7B-Instruct-GRPO | 67.55 | 504 | 59.52 | 378 | 72.30 | 396 | 66.12 | 535 | 48.46 | 575 | 14.48 | 592 | 54.74 | 497 | +0.11 |
| Kimi1.5 (Team et al., 2025) | **69.99** | 285 | 58.95 | 225 | 75.00 | 236 | **66.71** | 338 | 46.92 | 376 | 14.89 | 330 | **55.41** | 298 | +0.55 |
| CosFn (Yeo et al., 2025) | 37.94 | 1007 | 49.29 | 919 | **75.10** | 885 | 23.32 | 998 | 32.82 | 1012 | 10.12 | 1020 | 38.10 | 974 | -2.38 |
| DAST (Shen et al., 2025) | 65.90 | 260 | **59.67** | 327 | 69.70 | 153 | 65.86 | 205 | 50.51 | 466 | **15.59** | 288 | 54.54 | 283 | +0.54 |
| ShortBetter (Yi et al., 2025) | 67.94 | 333 | 58.90 | 250 | 71.70 | 260 | 65.86 | 376 | 46.92 | 405 | 14.36 | 387 | 54.28 | 335 | +0.41 |
| **AdaThink-Med-Qwen** | 68.34 | **79** | 58.74 | **109** | 73.50 | **62** | 66.45 | **121** | 48.72 | **175** | 14.16 | **88** | 54.99 | **106** | **+0.93** |

emergent property—rather, it is a direct consequence of our difficulty-aware length reward, which jointly penalizes verbosity on easy-correct samples and rewards exploration on hard-incorrect ones. The contribution, instead, is that this mode separation arises *end-to-end within a single model*, whereas prior work (Zhang et al., 2025) relies on manually annotated data and an external difficulty classifier to route between two separately trained models. To our knowledge, this is the first demonstration in medical QA that such bi-modal reasoning behavior can be learned end-to-end without auxiliary supervision.

**Dataset Selection Application.** Dataset selection (Albalak et al., 2024) aims to identify high-quality subsets that enable models to achieve comparable or superior performance with reduced training data. We leverage the intrinsic property of AdaThink-Med, where output length effectively correlates with problem difficulty, to filter high-value samples for

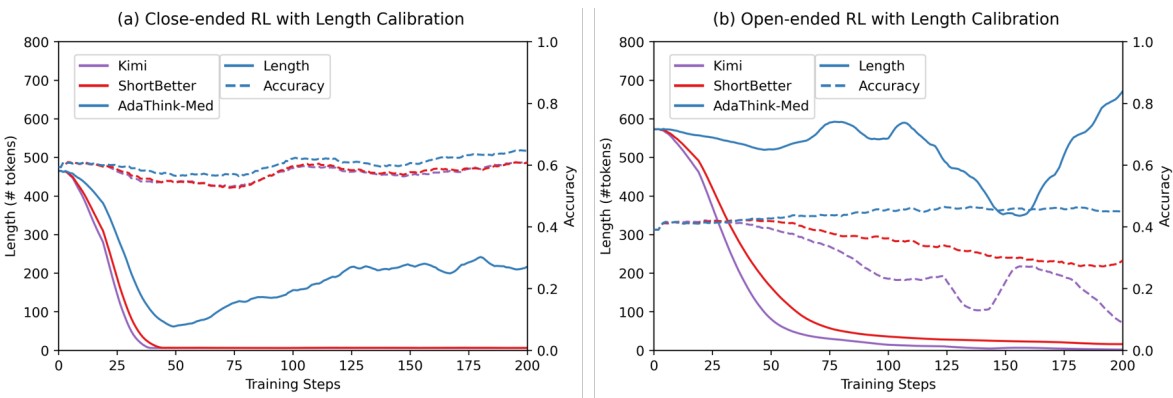

*Figure 2.* Length-reward hack during RL with greedy length calibration. Within each panel, accuracy (left $y$-axis) and average output length (right $y$-axis) are plotted against training step. **(a)** Closed-ended RL setup: both Kimi1.5 and ShortBetter collapse—accuracy and length drop sharply and do not recover, while AdaThink-Med (with performance compensation) preserves accuracy while compressing length. **(b)** Open-ended RL setup (following HuatuoGPT-o1): collapse persists even without candidate options, confirming that the failure is not an artifact of multiple-choice shortcuts. AdaThink-Med again maintains stable accuracy throughout training. For visual clarity, accuracy and length curves use distinct colors and markers.

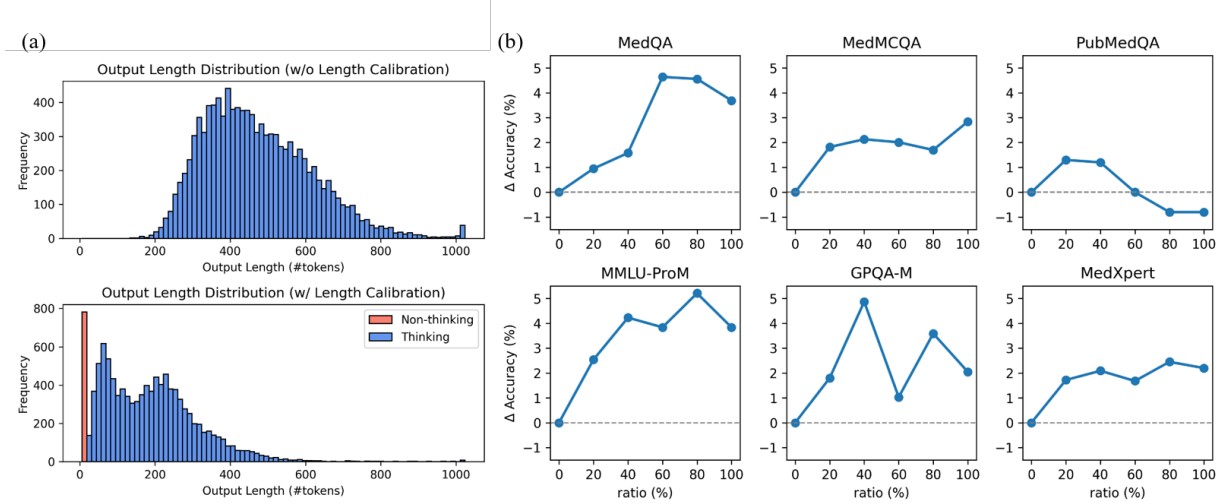

*Figure 3.* (a) Output length distribution w and w/o length calibration. (b) Dataset selection results.

reinforcement learning. Methodologically, we analyze the AlphaMed19k dataset using AdaThink-Med-Qwen to derive an output length distribution, subsequently partitioning samples into easy' (short) and hard' (long) categories based on the median length. We then construct stratified subsets at retention ratios of 20%, 40%, 60%, and 80% to investigate optimal data efficiency. As illustrated in Fig. 3(b), a 40% retention ratio not only matches but exceeds full-dataset performance, yielding gains of 2.10% on PubMedQA and 2.82% on GPQA-M. Remarkably, even at 20% retention, the model preserves 98.8% of the full-set performance. Detailed control experiments demonstrating the superiority of this strategy over random subset selection are provided in the Appendix, confirming that our method selects intrinsically higher-quality training instances.

**Human Evaluation: Clinical Validity** To assess whether the efficiency gains compromise clinical rigor, we conducted a blind human evaluation involving three board-certified Internal Medicine physicians (avg. 12 years of clinical experience; Fleiss' $\kappa = 0.76$, indicating substantial agreement; disagreements resolved via consensus) on 500 randomly sampled test cases, measuring *Response Accuracy*, *Reasoning Sufficiency* (the retention of essential logical steps), and *Logical Soundness* (reasoning chains free from hallucinations or flawed deductions). As detailed in Table 4, aggressive length-reduction baselines exhibit "shortcut hallucinations"—producing concise but medically flawed rationales to exploit brevity rewards, resulting in Logical Soundness below 70%. In contrast, AdaThink-Med achieves the best results on all three axes (Accuracy 76.25%, Sufficiency 89.00%, Logical Soundness 86.40%), demon-

*Table 3.* Ablation study on $\tau$ and $\alpha$ across in-domain and out-of-domain datasets. We rewrite the result of $\tau = 0.7, \alpha = 0.5$ for clear comparison.

| $\tau$ | $\alpha$ | MedQA | | MedMCQA | | PubMedQA | | MMLU-ProM | | GPQA-M | | MedXpert | | Avg | | AES |
|---|---|---|---|---|---|---|---|---|---|---|---|---|---|---|---|---|
| | | Acc. | Len. | Acc. | Len. | Acc. | Len. | Acc. | Len. | Acc. | Len. | Acc. | Len. | Acc. | Len. | |
| 0.3 | 0.5 | 68.19 | 391 | 58.76 | 307 | 73.90 | 311 | 68.08 | 434 | 50 | 483 | 14.37 | 457 | 55.55 | 397 | 0.36 |
| 0.5 | 0.5 | 67.32 | 105 | 59.12 | 170 | 72.80 | 84 | 67.17 | 196 | 47.69 | 245 | 15.02 | 116 | 54.85 | 153 | 0.82 |
| 0.7 | 0.5 | 68.34 | 79 | 58.74 | 109 | 73.50 | 62 | 66.45 | 121 | 48.72 | 175 | 14.16 | 88 | 54.99 | 106 | 0.93 |
| 0.9 | 0.5 | 62.53 | 7 | 58.50 | 13 | 74.00 | 6 | 64.30 | 11 | 46.67 | 12 | 14.49 | 8 | 53.41 | 10 | 1.04 |
| 0.7 | 0.1 | 66.06 | 121 | 59.02 | 153 | 74.30 | 116 | 66.19 | 195 | 50.51 | 258 | 15.39 | 142 | 55.25 | 164 | 0.82 |
| 0.7 | 0.5 | 68.34 | 79 | 58.74 | 109 | 73.50 | 62 | 66.45 | 121 | 48.72 | 175 | 14.16 | 88 | 54.99 | 106 | 0.93 |
| 0.7 | 0.9 | 63.47 | 44 | 58.12 | 216 | 73.60 | 10 | 63.19 | 50 | 46.92 | 69 | 14.08 | 30 | 53.23 | 70 | 0.90 |

*Table 4.* Human evaluation results on 500 randomly sampled test cases by three board-certified Internal Medicine physicians (Fleiss' $\kappa = 0.76$). **Reasoning Sufficiency** measures retention of essential logical steps; **Logical Soundness** measures whether the reasoning chain is free from hallucinations or flawed deductions. AdaThink-Med achieves the best balance across all three axes.

| Method | Accuracy (%) | Reasoning Sufficiency (%) | Logical Soundness (%) |
|---|---|---|---|
| Kimi | 71.25 | 73.00 | 68.40 |
| CosFn | 70.34 | 67.00 | 65.20 |
| DAST | 68.37 | 65.00 | 62.80 |
| ShortBetter | 70.23 | 72.00 | 69.50 |
| **AdaThink-Med** | **76.25** | **89.00** | **86.40** |

*Table 5.* **Open-ended evaluation on 796 real-world cardiac stroke cases**, scored blindly by three board-certified clinicians. *Diag.* = Diagnosis; *Treat.* = Treatment Plan; *Red.* ↓ lower is better; *ICA* = Clinical logical soundness and completeness.

| Method | Diag. Acc. | Diag. Red. ↓ | Diag. ICA | Treat. Acc. | Treat. Red. ↓ | Treat. ICA |
|---|---|---|---|---|---|---|
| Standard GRPO | 81.7 | 68.7 | 76.3 | 74.5 | 75.1 | 72.9 |
| Kimi | 78.3 | 35.2 | 71.3 | 70.5 | 38.4 | 64.2 |
| CosFn | 77.2 | 42.1 | 68.5 | 69.2 | 45.3 | 62.5 |
| DAST | 76.6 | 38.2 | 66.7 | 68.2 | 40.8 | 60.7 |
| ShortBetter | 77.2 | 25.4 | 69.4 | 68.7 | 22.1 | 61.5 |
| **AdaThink-Med** | **83.2** | **18.6** | **78.5** | **77.8** | **24.3** | **74.2** |

strating that our uncertainty-guided mechanism only compresses reasoning when the model is both correct and confident, preserving the fidelity of clinical decision-making.

**Real-World Open-Ended Clinical Evaluation.** To validate clinical utility beyond multiple-choice QA, we additionally conducted an open-ended evaluation on **796 anonymized real-world cardiac stroke cases** curated in collaboration with four hospitals. Each model receives the patient's clinical record and must generate a free-text *Diagnosis* and a free-text *Treatment Plan*; three board-certified clinicians then blindly score every generation on *Accuracy*, *Redundancy* (lower is better), and *Clinical ICA* (logical soundness and completeness). As shown in Table 5, AdaThink-Med attains the best Accuracy and Clinical ICA on both Diagnosis and Treatment, while keeping Redundancy at 18.6%/24.3%. Greedy length-reduction baselines (Kimi, ShortBetter) further reduce redundancy but at the cost of Clinical ICA dropping below 70%, indicating systematic over-pruning of clinically essential reasoning. This directly evidences, in a clinically realistic free-text generation setting, that our uncertainty-guided mechanism preserves diagnostic rigor while removing genuinely redundant content.

### 4.2. Ablations

**Impact of Problem Difficulty $\tau$ and Uncertainty Weight $\alpha$.** Since the parameter $\tau$ coarsely determines the overall compression ratio of the response length, we first fix the difficulty metric by setting the weight $\alpha$ for the uncertainty

measure to 0.5, in order to investigate the impact of the difficulty threshold $\tau$. A grid-search over $\{0.3, 0.5, 0.7, 0.9\}$ is conducted for the optimal $\tau$. As shown in Table 3, when $\tau$ is set too high (0.9), the average length on the test set is substantially reduced along with a degradation in performance and length reward hack. Conversely, $\tau = 0.3$ produces longer reasoning chains across the dataset (redundant overthinking), even though performance improves slightly. Next, we examine two values for $\alpha$, i.e., 0.1 and 0.9, while fixing $\tau$. Table 3 shows that $\alpha = 0.1$ (accuracy-focused) gives weaker performance (AES 0.82), while $\alpha = 0.9$ (uncertainty-focused) leads to only a slight drop (AES 0.90), highlighting the benefit of uncertainty-based difficulty for efficient reasoning. $\alpha = 0.5$ exhibits the best performance among the candidates. We believe an over-reliance on the uncertainty measure ($\alpha = 0.9$) may lead to a lag in performance compensation as its estimation precision increases. Overall, $\alpha = 0.5$ is identified as the optimal setting under the balanced $\tau = 0.7$.

**Effectiveness of Staged Adaptive Training.** In the initial phase of reinforcement learning, the model often exhibits low prediction accuracy and significant uncertainty, causing most questions to be perceived as hard. Emphasizing samples with incorrect outputs, our model inherently prioritizes accuracy improvement, leading to an imbalance between reducing overthinking and enhancing overall performance. To address this, we introduce a staged adaptive training strategy: the base model is first trained to convergence without length calibration, after which adaptive training is applied. As illustrated in Fig. 4, omitting the staged approach leads

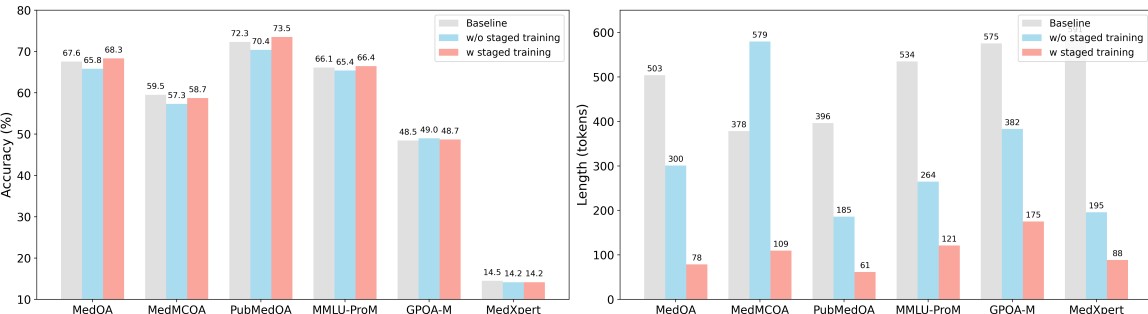

*Figure 4.* Performance comparison w and w/o staged adaptive training.

to limited length compression and noticeable performance degradation. In contrast, staged training maintains performance while achieving greater reduction in output length, demonstrating its effectiveness.

## 5. Conclusion

In this paper, we propose **AdaThink-Med**, an Uncertainty-Guided adaptive thinking framework designed to improve the efficiency of medical large language models. By dynamically shaping output lengths based on problem difficulty, AdaThink-Med effectively reduces redundant reasoning while maintaining competitive accuracy. Furthermore, we propose a data selection strategy that identifies high-quality reasoning and non-reasoning samples, enabling efficient training with significantly fewer data. Extensive experiments across six medical reasoning benchmarks demonstrate that AdaThink-Med achieves state-of-the-art overall performance under the trade-off between efficiency and accuracy, and produces a clear "thinking" vs. "non-thinking" mode separation within a single model as a direct consequence of our reward design, eliminating the need for external routers or classifiers used by prior pipeline approaches. We further discuss key limitations and promising future directions in Appendix 5.

## Limitations and Future Work

While AdaThink-Med achieves strong empirical efficiency gains, several limitations should be acknowledged. (i) **Reliance on entropy quality.** Our difficulty estimator depends on output entropy derived from model-generated rollouts; if the model is severely miscalibrated or rollouts are highly homogeneous, entropy may become a weak proxy for true difficulty. Although we observed that intra-batch min-max normalization and EMA smoothing already make our pipeline robust in practice (Appendix G.6), more principled uncertainty estimators (e.g., learned calibration heads or process reward models) deserve further investigation. (ii) **Step-level verification.** The current framework operates at the sequence level. Integrating step-level verifiers such as

Process Reward Models (PRMs) could provide finer-grained signals on which reasoning step is redundant versus essential, potentially improving Logical Soundness further. (iii) **Single-turn QA scope.** Our evaluation, while spanning six benchmarks and a real-world cardiac stroke study (Appendix G.1), is primarily single-turn. In multi-turn clinical workflows (e.g., iterative differential diagnosis, longitudinal consultations) or in long-context settings, reasoning traces accumulate over multiple turns and significantly degrade both performance and cost; the ability of AdaThink-Med to keep reasoning concise on simple turns is expected to be particularly beneficial in such regimes, and represents a natural direction for future deployment studies. (iv) **Cross-domain transfer.** Although the core mechanism is domain-agnostic, calibration of $\tau$ and $\alpha$ may need to be re-tuned when transferring across specialties, languages, or patient populations. Systematic study of cross-domain transfer of difficulty thresholds is left to future work. (v) **Backbone-specific stabilization.** As detailed in Appendix G.2, Llama backbones additionally require an $n$-gram repetition penalty to prevent reward hacking via degenerate repetition; this fragility motivates further work on backbone-agnostic stabilization techniques.

## Acknowledgments

This work was supported by the Shanghai Innovation Institute.

## Impact Statement

By dynamically optimizing inference compute based on clinical complexity, AdaThink-Med reduces the latency, cost, and energy footprint of medical reasoning models in clinical workflows. We emphasize, however, that AdaThink-Med is a clinical *decision-support* tool requiring rigorous prospective validation and human-in-the-loop oversight before deployment; calibration may also need to be re-validated when transferring across patient populations, languages, or specialties.

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

## Appendix Contents

# A. Reproducibility Statement

We place strong emphasis on reproducibility in this work. Detailed descriptions of experimental configurations—including model architectures, hyperparameter settings, and training strategies—are provided in Appendix E.1, with further implementation specifics available in Appendix E.2. To further support transparent validation, the complete source code and step-by-step instructions for replicating all experiments are publicly available at `https://github.com/shaohao011/AdaThinkMed`.

# B. More Related Works

**Medical LLMs.** With rapid advancements in training data and synthetic data formulation, LLMs in the medical domain have achieved significant performance improvements. Notable developments include instruction-tuned medical chatbot series (Ankit Pal, 2024; Qiu et al., 2024; Christophe et al., 2024; Zhang et al., 2024), reasoning models incorporating Chain-of-Thought (CoT) methodologies (Chen et al., 2024b; Huang et al., 2025; Rui et al., 2026), and agent collaboration (Chen et al., 2025a). These models have seen substantial adoption in both academic research and real-world applications. However, for clinical deployment, computational efficiency remains a critical concern alongside diagnostic accuracy. While SynapseRoute (Zhang et al., 2025) explores a router network to partition LLMs into "thinking" and "non-thinking" pathways, this approach requires additional manual labeling to train difficulty-classification networks, incurring extra computational and human resource costs. In contrast, we propose an efficient method that allocates computational resources within a single model without requiring supplementary modules or human intervention, thereby expanding potential application domains.

**Difficulty-Aware Length Control.** A concurrent line of work directly targets difficulty-aware reasoning length control. Selective Chain-of-Thought for medical QA (Zhan et al., 2026) introduces an explicit routing classifier that decides at inference whether to invoke a long CoT, requiring a separately trained router. L1 (Aggarwal & Welleck, 2025) learns a length controller jointly with the policy via reinforcement learning, but uses a static correctness signal and does not model predictive uncertainty. The Overthinker's DIET (Chen et al., 2026) uses static problem-difficulty estimates obtained offline and applies fixed length budgets per difficulty bucket. Compared with these methods, AdaThink-Med (i) operates end-to-end within a single model with *no* external router or classifier; (ii) couples *dynamic* rollout correctness with *predictive uncertainty* (Top-$K$ token entropy) inside the RL reward, making the difficulty signal co-evolve with the policy; and (iii) is tailored to the extreme complexity variance of clinical queries, where length budgets cannot be statically pre-assigned per question.

**Detailed Differentiation from Entropy-Guided Methods.** We further detail the differences between AdaThink-Med and three representative entropy-based reasoning techniques. (1) *vs. SEED-GRPO* (Chen et al., 2025c): SEED-GRPO uses semantic entropy to modulate the magnitude of policy updates (smaller gradients for uncertain queries) but leaves the reward function unchanged. AdaThink-Med fuses Top-$K$ token entropy with rollout correctness to *redesign the RL reward*, directly teaching the model to adjust generation length based on difficulty. (2) *vs. Adaptive Termination* (Xu et al., 2025): This is a strict test-time intervention that externally halts parallel reasoning branches once a semantic entropy threshold is crossed. AdaThink-Med is a *training-time* solution—length planning is internalized into the model's weights via RL, enabling autonomous mode switching in a single pass without an external controller. (3) *vs. Step-Entropy Compression* (Li et al., 2025): This inserts `[SKIP]` tokens to prune low-entropy reasoning steps; rigid pruning can break reasoning on hard problems. AdaThink-Med operates at the *sequence level*, fusing uncertainty with correctness into a Difficulty Score that provides "performance compensation"—compressing easy queries while *incentivizing extended reasoning* on hard cases, avoiding accuracy collapse from rigid pruning.

# C. Problem Formulation

Efficient reasoning aims to optimize a reasoning model $\pi_\theta$ parameterized by $\theta$ to generate high-quality outputs with reduced inference cost. Given a dataset $\mathcal{D}$, where each sample consists of an input prompt $\mathbf{x}$ and the corresponding ground-truth answer $\mathbf{y}^*$, the model produces an output $\mathbf{y} \sim \pi_\theta(\cdot|\mathbf{x})$. A reward function $R(\mathbf{y}, \mathbf{y}^*)$ is defined to evaluate the correctness of the generated output $\mathbf{y}$. To ensure stable optimization and prevent overfitting, a reference model $\pi_{\text{ref}}$ is introduced to regularize the policy update via a KL-divergence penalty term. The optimization objective for efficient reasoning is defined as:

$$\max_\theta \mathbb{E}_{\mathbf{x}\sim\mathcal{D},\mathbf{y}\sim\pi_\theta(\cdot|\mathbf{x})}\Big[R(\mathbf{y}, \mathbf{y}^*) - \lambda \cdot \mathcal{C}(\pi_\theta, \mathbf{x}) - \beta \cdot D_{\text{KL}}\big(\pi_\theta(\mathbf{y}|\mathbf{x}) \,\|\, \pi_{\text{ref}}(\mathbf{y}|\mathbf{x})\big)\Big] \tag{7}$$

where $\mathcal{C}(\pi_\theta, \mathbf{x})$ denotes the computational cost of generating $\mathbf{y}$ for the input $\mathbf{x}$, $\lambda$ controls the trade-off between reward and computational efficiency, and $\beta$ controls the influence of the KL-divergence regularization term. The reference model $\pi_{\text{ref}}$ corresponds to the model state prior to efficient reasoning optimization. We use GRPO as our reinforcement learning method and the details can be found at Appendix D.

## D. RL Algorithm

**Group Relative Policy Optimization (GRPO).** To optimize the efficient reasoning objective in Eq. (7), both critic-based reinforcement learning methods (e.g., PPO) and critic-free methods (e.g., GRPO (Guo et al., 2024)) can be applied. Considering its simplicity and effectiveness, GRPO has been widely adopted in recent works aiming to enhance the reasoning ability of large language models. Given an input prompt $\mathbf{x}$, a group of $G$ candidate outputs $\{o_i\}_{i=1}^G$ is sampled from the current policy $\pi_\theta$, where each $o_i = \{o_{i,1}, \ldots, o_{i,|o_i|}\}$ corresponds to one possible output sequence $\mathbf{y}$. We reformulate the objective of maximizing the expected reward $R(\mathbf{y}, \mathbf{y}^*)$ in Eq. (7) as minimizing the following GRPO loss:

$$
\begin{aligned}
\mathcal{L}_{\text{GRPO}}(\theta) = -\frac{1}{\sum_{i=1}^G |o_i|} \sum_{i=1}^G \sum_{t=1}^{|o_i|} \min \Bigg( \\
\frac{\pi_\theta(o_{i,t} \mid \mathbf{x}, o_{i,<t})}{\pi_{\theta_{\text{old}}}(o_{i,t} \mid \mathbf{x}, o_{i,<t})} \hat{A}_{i,t}, \\
\text{clip} \left( \frac{\pi_\theta(o_{i,t} \mid \mathbf{x}, o_{i,<t})}{\pi_{\theta_{\text{old}}}(o_{i,t} \mid \mathbf{x}, o_{i,<t})}, 1 - \epsilon, 1 + \epsilon \right) \hat{A}_{i,t} \Bigg).
\end{aligned}
\tag{8}
$$

where $\hat{A}_{i,t}$ denotes the relative advantage of token $o_{i,t}$ within the group, $\pi_{\theta_{\text{old}}}$ represents the policy before the update.

## E. AdaThink-Med Evaluation Details

### E.1. Baseline Models

We compare AdaThink-Med with strong large language models from both general-purpose and medical domains to enable a comprehensive evaluation. The general-purpose baselines are Qwen-2.5-Instruct-7B and Llama-3.1-Instruct-8B. The medical-domain baselines include MedLlama3, MMed, Med42 (Christophe et al., 2024), OpenBioLLM (Ankit Pal, 2024), UltraMedical (Zhang et al., 2024), HuatuoGPT-o1 (Chen et al., 2024b), and m1 (Huang et al., 2025). Length calibration RL based baselines include representative Kimi1.5 (Team et al., 2025), ShortBetter (Yi et al., 2025),DAST (Shen et al., 2025)and CosFn (Yeo et al., 2025). To counteract the length reward hack issue observed in models like Kimi1.5 and ShortBetter, we select the checkpoint from the stable performance period preceding the point where accuracy declines by more than 10% from its peak.

### E.2. Implementation Details

**Basic Setups.** Training is conducted using $8 \times$ H100 GPUs (80GB VRAM each) with PyTorch, leveraging FlashAttention-2 for computational efficiency. GRPO-related experiments are implemented using the `verl` (Sheng et al., 2024) framework to accelerate the training. We adopt the Qwen2.5 (Team, 2024)-7B-Instruct model and Llama3.1-Instruct-8B as the backbone models. The training uses a total batch size of 256, with a constant learning rate of $1e^{-6}$. The KL penalty coefficient is set to $\beta = 0.01$. For inference during GRPO, we deploy the model using `vllm` (Kwon et al., 2023) on 2 GPUs, generating 8 completions per sample, corresponding to the group size $G$ in GRPO. The temperature for `vllm` sampling is set to 1.0. As for our staged adaptive training, we first train the model using GRPO for 300 steps until full convergence is achieved. Subsequently, the proposed length calibration method is applied and train for 200 steps.

**Computational Overhead of the Difficulty Estimator.** The multiple rollouts required by our difficulty estimator are *inherent* to GRPO (which, being critic-free, already samples $G=8$ rollouts per prompt to compute relative advantages); unlike PPO, no separate critic network is needed. Our additional cost is therefore limited to (i) reading already-materialized token logits, (ii) computing Top-$K$ Shannon entropy, and (iii) a batch-level min-max normalization plus EMA threshold update. In wall-clock terms this adds **less than 1% overhead** per step on top of vanilla GRPO, since no extra forward passes are needed.

**Repetition Penalty with N-gram on LLama- Models**. We also observe that the model exhibits length reward hacking,

wherein it increases the length of reasoning chains on incorrectly answered hard questions through repetition rather than substantive reasoning. This finding aligns with the observations in Yeo et al. (2025). However, such length reward hacking is not observed in Qwen backbones, a discrepancy that warrants further investigation. In our experiments with LLaMA backbones, we apply the same n-gram repetition penalty as described in their work.

### E.3. Dataset Details

We conduct experiments on six publicly accessible medical and biomedical benchmarks, covering different knowledge domains and levels of reasoning difficulty. The datasets are summarized as follows:

**MedQA (Jin et al., 2021).** This benchmark is derived from the United States medical licensing examination and is designed to test comprehensive clinical knowledge across a wide range of medical specialties. The standard test split is used for evaluation.

**MedMCQA (Pal et al., 2022).** Originating from Indian medical entrance examinations, this dataset consists of multiple-choice questions that focus on fundamental medical concepts. Evaluation is performed on the official test set provided by the dataset.

**PubMedQA (Jin et al., 2019).** A biomedical question answering dataset constructed from PubMed abstracts. Each sample requires choosing between "yes," "no," or "maybe" to reflect factual understanding in biomedical research literature. We adopt the official test partition for all experiments.

**MMLU-ProM (Wang et al., 2024).** This is the medical subset of the general MMLU benchmark, specifically targeting professional-level medical and health-related knowledge. Following prior work (Chen et al., 2024b), we employ the standard split configuration defined for this subset.

**GPQA-M (Rein et al., 2023).** This dataset represents the biomedical branch of the Graduate-Level Question Answering benchmark. Its questions are manually curated to avoid superficial retrieval and to encourage deeper reasoning. We report results using the official evaluation split.

**MedXpert (Zuo et al., 2025).** A challenging benchmark built to assess advanced medical knowledge, complex reasoning, and clinical decision-making skills. It contains board-style exam questions across multiple body systems and specialties, curated by domain experts to ensure reliability and difficulty.

## F. Additional Experimental Results

### F.1. Design Rationale: Zero-Reward Quadrants

The piecewise length reward (Eq. 5) intentionally assigns *zero* length signal to two quadrants: EASY+INCORRECT ($\mathcal{D}_q < \theta_\mathcal{B} \land o_i \neq y^*$) and HARD+CORRECT ($\mathcal{D}_q > \theta_\mathcal{B} \land o_i = y^*$). This is a deliberate design choice that prioritizes accuracy over efficiency, justified by two observations from our early pilots with a symmetric 4-case reward:

- **Hard+Correct.** Applying a brevity penalty in this quadrant caused aggressive over-pruning of essential diagnostic reasoning chains and catastrophic accuracy drops on complex datasets such as GPQA-M. Zeroing the brevity penalty here *protects* fragile but successful reasoning chains, allowing the policy to consolidate them before being asked to compress.

- **Easy+Incorrect.** Penalizing length here discourages self-correction; rewarding length encourages hallucinated padding. Zeroing the length signal lets the missed accuracy reward $\mathcal{R}_{\text{acc}}=0$ dominate the gradient, forcing the model to first *fix the factual error* before any length-shaping is applied.

Empirically, this asymmetric design is what enables AdaThink-Med to compress aggressively ($4.7$–$6.4\times$) *without* the collapse modes observed in greedy length-calibration baselines (Kimi1.5, ShortBetter).

### F.2. Empirical Distribution of Difficulty Quadrants

We further characterize the four (correctness, uncertainty) quadrants induced by Eq. 3 to clarify how often each regime is encountered during training and to address the concern that "correct-but-uncertain" cases (*Lucky Correct*) might be misclassified as hard. Table 6 reports the empirical proportions during the initial phase of adaptive RL on AlphaMed19k. The LUCKY CORRECT regime accounts for only 14.4% initially and shrinks to ∼6% as training progresses. Crucially, in

*Table 6.* Empirical distribution of the four (correctness, uncertainty) quadrants at the start of adaptive RL. The "Lucky Correct" regime represents only ∼14% of samples and progressively shrinks to ∼6% as training stabilizes.

| Regime | Correctness | Uncertainty | Proportion |
|---|---|---|---|
| True Easy | Correct | Low | 36.3% |
| True Hard | Incorrect | High | 30.8% |
| Confident Error | Incorrect | Low | 18.5% |
| Lucky Correct | Correct | High | 14.4% |

ablations that *do* penalize length in this regime, the model learns truncated unreasoned guesses, degrading accuracy on related test questions. Withholding the length penalty allows the necessary token budget for the model to build rigorous evidence chains, gradually transitioning "lucky guesses" into "confident deductions."

### F.3. Per-Benchmark Difficulty Scores

Although $\theta_{\mathcal{B}}$ is computed globally over the training distribution (AlphaMed19k), we verify that the induced difficulty estimator generalizes meaningfully to out-of-domain (OOD) benchmarks. Table 7 reports per-benchmark average difficulty scores $\bar{\mathcal{D}}_q$ obtained by running the trained AdaThink-Med-Qwen offline on each test set. The ordering matches both expert

*Table 7.* Per-benchmark average difficulty scores $\bar{\mathcal{D}}_q$ from AdaThink-Med-Qwen. The model correctly ranks PubMedQA as easiest and MedXpert/GPQA-M as hardest, even for OOD benchmarks unseen during training.

| Benchmark | Split | Avg. $\bar{\mathcal{D}}_q$ |
|---|---|---|
| PubMedQA | OOD | 0.27 |
| MedQA | In-domain | 0.31 |
| MMLU-ProM | OOD | 0.46 |
| MedMCQA | In-domain | 0.52 |
| GPQA-M | OOD | 0.63 |
| MedXpert | OOD | 0.79 |

intuition and downstream accuracy: PubMedQA (factoid yes/no/maybe) is identified as the easiest, while GPQA-M and MedXpert—which contain expert-level clinical reasoning—are correctly identified as the hardest. This indicates that a single global threshold trained on AlphaMed19k transfers to OOD difficulty estimation without per-benchmark re-tuning.

### F.4. Dynamic Threshold Evolution During Training

Because $\theta_{\mathcal{B}}$ is the $\tau$-quantile of intra-batch difficulty scores under EMA smoothing, it acts as a *relative* (rather than absolute) boundary that adapts as the model improves. Empirically we observe a two-phase trajectory: in the first half of adaptive RL, the threshold exhibits a *downward drift*—as accuracy improves and entropy drops on previously hard examples, batch-wise $\mathcal{D}_q$ values decrease, lowering the quantile. In the second half, after the policy stabilizes, the threshold *converges* to a stable value. The EMA coefficient $\gamma$ damps batch-to-batch variance and prevents abrupt reward flips during this drift. This dynamic relative behavior is precisely what allows the estimator to keep tracking the model's evolving capability rather than freezing to an outdated notion of difficulty.

### F.5. Extended Ablation on the Uncertainty Weight $\alpha$

For completeness we extend the $\alpha$ ablation in Table 3 to include the two extreme corners $\alpha=0.0$ (pure correctness) and $\alpha=1.0$ (pure entropy), addressing the concern that "$\alpha=0.1$" in the main table is not strictly correctness-only. Results in Table 8 confirm the trend: at $\alpha=0.0$ the difficulty signal is purely based on correctness, which under $G=8$ rollouts gives only coarse 12.5% increments and the longest outputs (Avg. Len. 196, AES 0.76); at $\alpha=1.0$, removing the correctness anchor leads to the largest accuracy drop (Avg. Acc. 52.65%) because the model can no longer distinguish lucky guesses from true mastery. The balanced $\alpha=0.5$ remains the best operating point (AES 0.93).

*Table 8.* Extended $\alpha$ ablation including the extreme corners $\alpha$=0.0 and $\alpha$=1.0. The balanced $\alpha$=0.5 achieves the best AES.

| $\alpha$ | MedQA | | MedMCQA | | PubMedQA | | MMLU-ProM | | GPQA-M | | MedXpert | | Avg | | AES |
|---|---|---|---|---|---|---|---|---|---|---|---|---|---|---|---|
| | Acc. | Len. | Acc. | Len. | Acc. | Len. | Acc. | Len. | Acc. | Len. | Acc. | Len. | Acc. | Len. | |
| 0.0 (correctness-only) | 65.74 | 137 | 59.21 | 184 | 74.20 | 137 | 66.35 | 271 | 50.86 | 284 | 15.66 | 165 | 55.34 | 196 | 0.76 |
| 0.1 (accuracy-focused) | 66.06 | 121 | 59.02 | 153 | 74.30 | 116 | 66.19 | 195 | 50.51 | 258 | 15.39 | 142 | 55.25 | 164 | 0.82 |
| 0.5 (balanced) | 68.34 | 79 | 58.74 | 109 | 73.50 | 62 | 66.45 | 121 | 48.72 | 175 | 14.16 | 88 | 54.99 | 106 | **0.93** |
| 0.9 (uncertainty-focused) | 63.47 | 44 | 58.12 | 216 | 73.60 | 10 | 63.19 | 50 | 46.92 | 69 | 14.08 | 30 | 53.23 | 70 | 0.90 |
| 1.0 (entropy-only) | 62.85 | 38 | 57.34 | 224 | 73.20 | 8 | 62.41 | 41 | 46.26 | 57 | 13.81 | 24 | 52.65 | 65 | 0.88 |

## F.6. Sensitivity Analysis of TOP-$K$

To assess the impact of the hyperparameter TOP-$K$, we conducted ablation experiments using the Qwen2.5-Instruct-7B model. The results shown in Table 9 indicate that variations in the TOP-$K$ value lead to stable efficiency improvements in AdaThink-Med. However, both very large and very small values of TOP-$K$ result in suboptimal efficiency gains. This occurs because smaller $K$ values reduce the number of sampled tokens, leading to inaccurate difficulty estimations, while larger $K$ values introduce an averaging effect that dilutes the model's entropy due to the presence of plain tokens.

*Table 9.* Ablation results on the TOP-$K$ hyperparameter for entropy calculation.

| $K$ | MedQA | | MedMCQA | | PubMedQA | | MMLU-ProM | | GPQA-M | | MedXpert | | Avg. | | AES |
|---|---|---|---|---|---|---|---|---|---|---|---|---|---|---|---|
| | Acc. | Len. | Acc. | Len. | Acc. | Len. | Acc. | Len. | Acc. | Len. | Acc. | Len. | Acc. | Len. | |
| - | 63.86 | 490 | 65.68 | 359 | 73.10 | 386 | 62.23 | 523 | 46.41 | 562 | 12.28 | 585 | 52.44 | 484 | - |
| 5 | 67.83 | 102 | 67.53 | 125 | 74.22 | 87 | 65.93 | 131 | 46.96 | 221 | 13.95 | 102 | 54.41 | 128 | +0.85 |
| 20 | 68.34 | 79 | 68.74 | 109 | 73.50 | 62 | 66.40 | 121 | 48.72 | 175 | 14.16 | 109 | 54.99 | 106 | +0.93 |
| 50 | 67.55 | 141 | 68.35 | 127 | 73.92 | 105 | 68.32 | 143 | 46.79 | 190 | 14.24 | 117 | 54.86 | 137 | +0.86 |

## F.7. Sensitivity Analysis of the AES Metric

To ensure a fair and consistent comparison with prior work in adaptive reasoning, we adopted the Accuracy-Efficiency Score (AES) using the standard hyperparameters ($\alpha = 1, \beta = 3, \gamma = 5$) as established in recent studies (Yi et al., 2025; Luo et al., 2025). This metric was designed to provide a comprehensive assessment of the trade-off between reducing computational costs and maintaining model performance.

However, to address potential concerns regarding the sensitivity of the AES metric to specific hyperparameter choices, we conducted a robustness analysis. We evaluated AdaThink-Med and baseline models under a diverse set of AES configurations, systematically varying the weights for length reduction ($\alpha$), accuracy gain ($\beta$), and accuracy loss penalty ($\gamma$).

The results, presented in Table 10, demonstrate that AdaThink-Med consistently achieves the highest AES across all tested hyperparameter combinations. Whether the metric configuration prioritizes strict accuracy preservation (high $\gamma$) or aggressive efficiency gains (high $\alpha$), our method maintains its superiority. These findings confirm that the reported performance advantage is intrinsic to the adaptive reasoning mechanism of AdaThink-Med and is robust to variations in the evaluation metric.

## F.8. Effectiveness of AdaThink-Med on Non-R1-Like Models

The adaptive thinking experiments in the main paper are performed on R1-like models, where reasoning capability is acquired through RL training with GRPO. To further validate our approach, we conduct additional experiments on other reasoning models. Our implementation follows the official setup of HuatuoGPT-o1 (Chen et al., 2024b), which employs a large language model (LLM) as a judgment mechanism. Specifically, we use HuatuoGPT-o1-7B (Chen et al., 2024b) as our base model, which is built upon the Qwen2.5-Instruct-7B backbone and trained in two stages: first through supervised fine-tuning (SFT) on 20,000 long-form CoT samples, followed by reinforcement learning via proximal policy optimization (PPO) on 20,000 verifiable medical question-answering samples. All 40,000 samples are converted into open-ended QA format and sourced from the training splits of MedQA-USMLE and MedMCQA. The 20,000 PPO samples are further utilized in our subsequent adaptive GRPO training. For verification, we adopt the distilled verifier released by HuatuoGPT-o1, which is trained using 20,000 scoring samples distilled from GPT-4o.

*Table 10.* Sensitivity analysis of the Accuracy-Efficiency Score (AES) under varying hyperparameter settings. $\Delta L$ and $\Delta A$ represent the relative reduction in length and change in accuracy, respectively. The standard configuration used in the main text is $\alpha = 1, \beta = 3, \gamma = 5$. AdaThink-Med consistently outperforms baselines across all weighting schemes. Best results are in **bold**.

| Model | Avg Acc. | Avg Len. | $\Delta L$ | $\Delta A$ | AES Configurations $(\alpha, \beta, \gamma)$ | | | | | |
| --- | --- | --- | --- | --- | --- | --- | --- | --- | --- | --- |
| | | | | | 1, 3, 5 | 1, 5, 5 | 1, 2, 3 | 1, 2, 10 | 0.5, 2, 1 | 0.1, 1, 2 |
| *Llama-3.1-8B Backbone* | | | | | | | | | | |
| Llama-3.1-8B-Instruct [CoT] | 54.25 | 425.27 | - | - | - | - | - | - | - | - |
| Llama-3.1-8B-Instruct | 48.93 | 254.38 | 0.40 | -0.10 | -0.09 | -0.09 | 0.11 | -0.58 | 0.10 | -0.16 |
| Llama-3.1-8B-Instruct [GRPO] | 56.72 | 409.88 | 0.04 | 0.05 | 0.17 | 0.26 | 0.13 | 0.17 | 0.11 | 0.05 |
| Kimi | 55.65 | 124.89 | 0.71 | 0.03 | 0.78 | 0.84 | 0.76 | 0.78 | 0.40 | 0.10 |
| CosFn | 38.10 | 973.64 | -1.29 | -0.30 | -2.78 | -2.78 | -2.18 | -4.27 | -0.94 | -0.72 |
| DAST | 52.71 | 51.37 | 0.88 | -0.03 | 0.74 | 0.74 | 0.79 | 0.60 | 0.41 | 0.03 |
| ShortBetter | 55.94 | 167.38 | 0.61 | 0.03 | 0.70 | 0.76 | 0.67 | 0.70 | 0.37 | 0.09 |
| **AdaThink-Med-Llama** | 55.59 | 63.81 | 0.85 | 0.02 | **0.92** | **0.97** | **0.90** | **0.92** | **0.47** | **0.11** |
| *Qwen2.5-7B Backbone* | | | | | | | | | | |
| Qwen-Instruct-7B [CoT] | 52.44 | 484.08 | - | - | - | - | - | - | - | - |
| Qwen-Instruct-7B | 48.04 | 278.24 | 0.43 | -0.08 | 0.01 | 0.01 | 0.17 | -0.41 | 0.13 | -0.13 |
| Qwen-Instruct-7B [GRPO] | 54.74 | 496.62 | -0.03 | 0.04 | 0.11 | 0.19 | 0.06 | 0.11 | 0.07 | 0.04 |
| Kimi | 55.41 | 298.09 | 0.38 | 0.06 | 0.55 | 0.67 | 0.50 | 0.55 | 0.31 | 0.10 |
| CosFn | 38.10 | 973.64 | -1.01 | -0.27 | -2.38 | -2.38 | -1.83 | -3.75 | -0.78 | -0.65 |
| DAST | 54.54 | 283.01 | 0.42 | 0.04 | 0.54 | 0.62 | 0.50 | 0.54 | 0.29 | 0.08 |
| ShortBetter | 54.28 | 335.19 | 0.31 | 0.04 | 0.41 | 0.48 | 0.38 | 0.41 | 0.22 | 0.07 |
| **AdaThink-Med-Qwen** | 54.99 | 105.71 | 0.78 | 0.05 | **0.93** | **1.02** | **0.88** | **0.93** | **0.49** | **0.13** |

*Table 11.* Experimental results on using adaptive thinking methods on non-R1 reasoning models, here we use HuatuoGPT-o1, which first SFT with CoT data and then RL with PPO.

| Model | MedQA | | MedMCQA | | PubMedQA | | MMLU-ProM | | GPQA-M | | MedXpert | | Avg | | AES |
| --- | --- | --- | --- | --- | --- | --- | --- | --- | --- | --- | --- | --- | --- | --- | --- |
| | Acc. | Len. | Acc. | Len. | Acc. | Len. | Acc. | Len. | Acc. | Len. | Acc. | Len. | Acc. | Len. | |
| HuatuoGPT-o1-8B | 76.35 | 568 | 62.82 | 445 | 79.80 | 445 | 63.71 | 521 | 54.35 | 585 | 17.06 | 601 | 59.02 | 527 | – |
| Kimi1.5 (Team et al., 2025) | 71.01 | 373 | 62.87 | 302 | 77.40 | 288 | 67.04 | 340 | 50.26 | 401 | 14.49 | 363 | 57.18 | 345 | 0.19 |
| CosFn (Yeo et al., 2025) | 68.42 | 260 | 62.63 | 126 | 77.30 | 159 | 65.60 | 220 | 48.97 | 187 | 14.24 | 246 | 56.20 | 200 | 0.38 |
| DAST (Shen et al., 2025) | 71.17 | 730 | 61.01 | 619 | 76.70 | 578 | 64.36 | 697 | 45.90 | 794 | 15.02 | 765 | 55.69 | 697 | -0.60 |
| ShortBetter (Yi et al., 2025) | 71.64 | 336 | 63.18 | 271 | 78.40 | 258 | 66.12 | 299 | 49.74 | 353 | 14.45 | 321 | 57.26 | 306 | 0.27 |
| **AdaThink-Med-HuatuoGPT-o1** | 72.74 | 183 | 63.54 | 192 | 76.60 | 72 | 66.96 | 204 | 50.77 | 344 | 13.88 | 221 | 57.41 | 203 | 0.48 |

We compare several RL-based length calibration methods, and the results are summarized in Table 11. Our findings demonstrate that AdaThink-Med achieves the highest efficiency improvement. Specifically, AdaThink-Med reduces the average output length by a factor of 2.6, from 527 to 203, while incurring only an average performance degradation of 1.59%. In comparison, the second-best method, CosFn, compresses the average output length from 527 to 200, but with a larger performance degradation of 2.78%. Overall, these results confirm the effectiveness of AdaThink-Med in enhancing the inference efficiency of diverse reasoning models.

### F.9. Estimated Difficulty vs. Response Length Analysis

Additionally, we have included the requested plot of response length vs. estimated difficulty in Fig. 5. The overall trend observed is as follows: for correct test samples, as the response length increases, the uncertainty also rises, indicating an increase in difficulty. For incorrect samples, both short and long response lengths show high difficulty, with short responses exhibiting a more pronounced rise in uncertainty. This suggests that more reasoning or cognitive effort is required in such cases. Notably, in the region of medium response lengths, both uncertainty measures level off, indicating that these problems are challenging to optimize using a single mode.

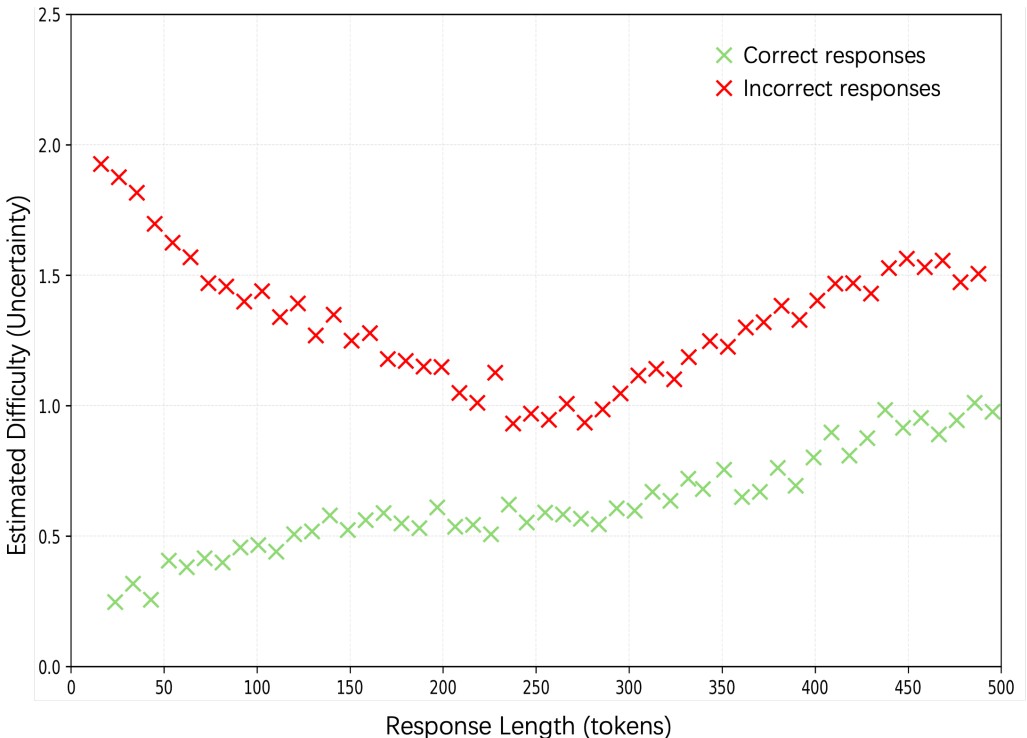

*Figure 5.* Estimated difficulty vs. response length.

## F.10. Generalization to Large-Scale Reasoning Models

To assess the generalizability of AdaThink-Med to larger-scale and state-of-the-art architectures, we extended our experimental evaluation to include Qwen-3-32B. As a reasoning-intensive model optimized through reinforcement learning, Qwen-3-32B serves as a robust benchmark for evaluating adaptive thinking capabilities in advanced LLMs. We directly applied the AdaThink-Med framework to this model without architectural modifications to test its adaptability.

The empirical results are summarized in Table 12. We observe that AdaThink-Med consistently achieves substantial reductions in inference length compared to the baseline and other length-control methods. Specifically, the average output length is reduced from 1888 tokens to 954 tokens, representing a compression of approximately 49%. Importantly, this efficiency gain comes with minimal performance degradation, resulting in the highest Accuracy-Efficiency Score (AES) of 0.45. These findings further corroborate the scalability and effectiveness of AdaThink-Med when applied to newer, larger parameter models that possess inherent reasoning capabilities.

*Table 12.* Performance evaluation on the large-scale Qwen-3-32B model. AdaThink-Med achieves superior efficiency gains while maintaining competitive accuracy across six medical benchmarks. The best AES scores are highlighted in **bold**.

| Model | MedQA | | MedMCQA | | PubMedQA | | MMLU-ProM | | GPQA-M | | MedXpert | | Avg | | AES |
|---|---|---|---|---|---|---|---|---|---|---|---|---|---|---|---|
| | Acc. | Len. | Acc. | Len. | Acc. | Len. | Acc. | Len. | Acc. | Len. | Acc. | Len. | Acc. | Len. | |
| Qwen3-32B | 87.50 | 1679 | 72.34 | 1473 | 73.50 | 722 | 79.67 | 1770 | 68.46 | 3009 | 26.41 | 2676 | 67.98 | 1888 | 0.00 |
| Kimi | 84.23 | 1099 | 69.93 | 1002 | 73.00 | 656 | 79.45 | 1335 | 67.25 | 1776 | 25.37 | 2108 | 66.54 | 1329 | +0.19 |
| CosFn | 85.60 | 1523 | 72.12 | 1355 | 72.20 | 585 | 78.53 | 1557 | 66.25 | 2553 | 24.45 | 2330 | 66.53 | 1651 | +0.02 |
| DAST | 86.53 | 1345 | 71.88 | 1154 | 73.10 | 724 | 76.53 | 1553 | 64.32 | 1889 | 24.32 | 2235 | 66.11 | 1513 | +0.06 |
| ShortBetter | 84.30 | 1427 | 70.15 | 1238 | 72.50 | 669 | 76.78 | 1476 | 67.24 | 1920 | 24.99 | 2140 | 65.99 | 1478 | +0.07 |
| AdaThink-Med | 86.20 | 1120 | 72.55 | 893 | 75.40 | 256 | 77.76 | 785 | 66.24 | 1431 | 25.88 | 1237 | 67.34 | 954 | **+0.45** |

# G. Further Analyses

## G.1. Real-World Open-Ended Clinical Evaluation: Additional Details

The open-ended evaluation on 796 anonymized real-world cardiac stroke cases is summarized in the main text (Section 4.1, Table 5). The dataset was curated in collaboration with four hospitals and de-identified prior to use. Each model received the raw clinical record (history of present illness, past medical history, examination, and relevant imaging notes) and was prompted to produce a free-text *Diagnosis* and a free-text *Treatment Plan*. Three board-certified clinicians blindly scored every generation along three axes: *Accuracy* (factual correctness against the gold standard), *Redundancy* (proportion of clinically non-essential content; lower is better), and *Clinical ICA* (logical soundness and completeness of the reasoning). Disagreements were resolved by consensus discussion. As reported in the main text, AdaThink-Med achieves the highest Accuracy and Clinical ICA on both subtasks while keeping Redundancy at 18.6% (Diagnosis) and 24.3% (Treatment). Greedy length-reduction baselines (Kimi, ShortBetter) further reduce Redundancy but at the cost of Clinical ICA dropping below 70%, indicating systematic over-pruning of clinically essential reasoning steps—a failure mode our uncertainty-guided mechanism explicitly prevents by tying compression to model confidence.

## G.2. Repetition Penalty Ablation on Llama

As briefly noted in Appendix E.2, training Llama-3.1-8B with adaptive length calibration requires an additional $n$-gram repetition penalty to prevent reward hacking via degenerate token repetition. Table 13 quantifies this: removing the repetition penalty causes *complete* training collapse rather than mild degradation. Without the penalty the policy enters degenerate $n$-gram loops, padding outputs to the maximum token budget without producing valid reasoning—consistent with prior observations on Llama backbones (Yeo et al., 2025). Qwen-2.5 backbones inherently resist this failure mode and do *not* require the penalty, so we apply it only on Llama for fairness across backbones.

*Table 13.* Effect of removing the $n$-gram repetition penalty on Llama-3.1-8B during adaptive RL. Average accuracy and length over the six medical benchmarks. Without the penalty, training collapses into degenerate repetition.

| Configuration | Avg. Acc. (%) | Avg. Len. | AES |
|---|---|---|---|
| Standard GRPO-Llama | 56.72 | 410 | $-0.09$ |
| AdaThink-Llama without repetition penalty | 13.26 | 1024 | $-3.08$ |
| AdaThink-Llama with repetition penalty | 55.59 | 64 | $+0.92$ |

## G.3. Controlled Compute-Matched GRPO Baseline

Our staged training pipeline first runs vanilla GRPO for 300 steps, then applies adaptive length calibration for an additional 200 steps (500 total). To rule out the possibility that AdaThink-Med's improvements over standard GRPO simply reflect more total compute, we run a compute-matched control: extend vanilla GRPO from 300 to 500 steps without any length calibration. Results in Table 14 show that GRPO(500) yields negligible additional improvement over GRPO(300) in both accuracy and length, confirming that the observed gains stem from the adaptive length reward design rather than from longer training.

*Table 14.* Compute-matched control: extending vanilla GRPO to the same total step count (500) as AdaThink-Med yields negligible change, confirming gains come from the reward design.

| Backbone | GRPO(300) Acc/Len | GRPO(500) Acc/Len |
|---|---|---|
| Llama-3.1-8B | 56.72% / 410 | 56.73% / 408 |
| Qwen-2.5-7B | 54.74% / 497 | 54.71% / 494 |
| **AdaThink-Med (Llama)** | 55.59% / 64 | |
| **AdaThink-Med (Qwen)** | 54.99% / 106 | |

*Table 15.* Experimental results showing the robustness of AdaThink-Med under low-quality output conditions.

| Method | PubMedQA | | MMLU-ProM | | GPQA-M | | MedXpert | | Avg. | |
|---|---|---|---|---|---|---|---|---|---|---|
| | Acc. | Len. | Acc. | Len. | Acc. | Len. | Acc. | Len. | Acc. | Len. |
| Qwen-GRPO | 72.30 | 396 | 66.12 | 535 | 48.46 | 575 | 14.48 | 592 | 50.34 | 524 |
| Kimi | 71.20 | 230 | 65.23 | 328 | 46.52 | 376 | 14.31 | 366 | 49.34 | 325 |
| CosFn | 70.90 | 301 | 64.17 | 428 | 47.66 | 351 | 13.87 | 353 | 49.15 | 358 |
| DAST | 72.30 | 298 | 64.88 | 535 | 47.35 | 507 | 13.26 | 498 | 49.45 | 460 |
| ShortBetter | 70.50 | 354 | 63.15 | 297 | 45.23 | 330 | 13.26 | 320 | 48.04 | 325 |
| AdaThink-Med | 72.10 | 89 | 65.38 | 115 | 47.51 | 212 | 14.33 | 108 | 49.83 | 131 |

## G.4. Robustness to Low-Quality Outputs

To assess the robustness of our algorithm to low-quality outputs, we conducted experiments using a low-quality dataset. Specifically, after training the Qwen2.5-Instruct-7B model with GRPO, we performed multiple rounds of response reasoning for the same question in the training set. We then ranked the samples based on accuracy and selected both the fully incorrect samples and some low-accuracy samples to form our low-quality dataset. These samples, which generate high uncertainty, present a significant challenge for training. As shown in Table 15, the experimental results indicate that AdaThink-Med consistently outperforms other models under these challenging conditions, demonstrating the robustness of our algorithm.

## G.5. VQA Generalization Ability

To evaluate the effectiveness of AdaThink-Med in multi-modal scenarios, we conducted a clinical free-text generation task. Specifically, we included experimental results from the radiology (X-ray) report generation task, which is a Visual Question Answering (VQA) task. The MIMIC-CXR-2.0 (Johnson et al., 2019) dataset was used for validation. In the test set, we employed LLM-as-judge to assess the generated reports and compare them to the ground truth in two key aspects: the completeness of the report and its conciseness. As shown in Table 17, the results demonstrate that AdaThink-Med effectively reduces the report length while maintaining completeness, highlighting its exceptional performance.

## G.6. Sensitivity Analysis of Entropy Estimation via Mixed LLMs

To evaluate the robustness of our entropy-based difficulty estimator and explore potential performance gains from model ensembling, we conducted a sensitivity analysis by utilizing a mixture of LLMs for entropy estimation. In our main experiments, the entropy used for difficulty estimation is derived from the target model itself (self-estimation). In this supplementary experiment, we introduce a cross-model estimation strategy.

Specifically, we define **AdaThink-Med-Llama(+Qwen)** as the Llama-3.1-8B-Instruct model where the adaptive reasoning process is guided by entropy signals derived from a GRPO-enhanced Qwen2.5-7B-Instruct model. Conversely, **AdaThink-Med-Qwen(+Llama)** utilizes the Llama model to estimate uncertainty for the Qwen backbone. This approach leverages the diversity of probability distributions across different LLM architectures to provide a more comprehensive and stable entropy estimate.

The experimental results are presented in Table 16. We observe that employing mixed LLMs for entropy estimation yields consistent improvements over the single-model baseline:

1. **Improved Accuracy and Efficiency:** The mixed-model approach results in more accurate difficulty estimation, allowing the model to prune redundant reasoning steps more aggressively without sacrificing correctness. For instance, AdaThink-Med-Llama(+Qwen) achieves an Accuracy-Efficiency Score (AES) of **1.06**, surpassing the single-model version (0.92).

2. **Mitigation of Performance Drops:** This strategy effectively addresses the performance fluctuations observed in specific datasets. Notably, on the MedQA dataset, where the single-model Llama variant experienced a slight accuracy trade-off, the Qwen-guided variant recovers performance significantly (from 67.00% to 74.21%) while further reducing the average output length (from 36 to 33 tokens).

These findings suggest that our uncertainty-guided length calibration mechanism is not only robust but can be further

enhanced through ensemble-based entropy estimation, paving the way for more efficient and accurate medical reasoning systems.

*Table 16.* Performance comparison of AdaThink-Med using single vs. mixed LLMs for entropy estimation across six medical benchmarks. "+Model" indicates the source of external entropy guidance. The best AES scores are highlighted in **bold**.

| Model | MedQA | | MedMCQA | | PubMedQA | | MMLU-ProM | | GPQA-M | | MedXpert | | Avg | | AES |
|---|---|---|---|---|---|---|---|---|---|---|---|---|---|---|---|
| | Acc. | Len. | Acc. | Len. | Acc. | Len. | Acc. | Len. | Acc. | Len. | Acc. | Len. | Acc. | Len. | |
| *Llama-3.1-8B Backbone* | | | | | | | | | | | | | | | |
| Llama-3.1-8B-Instruct [CoT] | 68.18 | 461 | 57.32 | 309 | 78.20 | 296 | 60.06 | 457 | 45.64 | 485 | 16.08 | 545 | 54.25 | 425 | - |
| Llama-3.1-8B-Instruct | 54.75 | 302 | 54.69 | 169 | 77.00 | 113 | 57.00 | 256 | 35.64 | 345 | 14.48 | 342 | 48.93 | 254 | -0.09 |
| Llama-3.1-8B-Instruct [GRPO] | 72.50 | 438 | 61.60 | 299 | 78.50 | 290 | 63.51 | 442 | 46.92 | 461 | 17.26 | 529 | 56.72 | 410 | 0.17 |
| Kimi | 70.62 | 145 | 60.93 | 100 | 77.50 | 92 | 63.71 | 164 | 45.38 | 118 | 15.75 | 131 | 55.65 | 125 | 0.78 |
| CosFn | 37.94 | 1007 | 49.29 | 919 | 75.10 | 885 | 23.32 | 998 | 32.82 | 1012 | 10.12 | 1020 | 38.10 | 974 | -2.78 |
| DAST | 60.64 | 30 | 56.90 | 34 | 76.20 | 39 | 59.48 | 71 | 47.69 | 53 | 15.35 | 82 | 52.71 | 51 | 0.74 |
| ShortBetter | 71.48 | 175 | 60.62 | 134 | 78.60 | 118 | 62.99 | 188 | 45.89 | 219 | 16.08 | 171 | 55.94 | 167 | 0.70 |
| AdaThink-Med-Llama | 67.00 | 36 | 60.41 | 73 | 78.40 | 54 | 63.58 | 85 | 47.94 | 100 | 16.20 | 35 | 55.59 | 64 | 0.92 |
| **AdaThink-Med-Llama(+Qwen)** | **74.21** | **33** | **62.35** | **60** | **78.22** | **32** | **63.59** | **78** | **48.21** | **88** | **18.77** | **30** | **57.56** | **54** | **1.06** |
| *Qwen2.5-7B Backbone* | | | | | | | | | | | | | | | |
| Qwen-Instruct-7B [CoT] | 63.86 | 490 | 56.68 | 359 | 73.10 | 386 | 62.28 | 523 | 46.41 | 562 | 12.28 | 585 | 52.44 | 484 | - |
| Qwen-Instruct-7B | 54.28 | 313 | 53.43 | 196 | 72.70 | 129 | 56.67 | 292 | 38.71 | 448 | 12.44 | 291 | 48.04 | 278 | 0.01 |
| Qwen-Instruct-7B [GRPO] | 67.55 | 504 | 59.52 | 378 | 72.30 | 396 | 66.12 | 535 | 48.46 | 575 | 14.48 | 592 | 54.74 | 497 | 0.11 |
| Kimi | 69.99 | 285 | 58.95 | 225 | 75.00 | 236 | 66.71 | 338 | 46.92 | 376 | 14.89 | 330 | 55.41 | 298 | 0.55 |
| CosFn | 37.94 | 1007 | 49.29 | 919 | 75.10 | 885 | 23.32 | 998 | 32.82 | 1012 | 10.12 | 1020 | 38.10 | 974 | -2.38 |
| DAST | 65.90 | 260 | 59.67 | 327 | 69.70 | 153 | 65.86 | 205 | 50.51 | 466 | 15.59 | 288 | 54.54 | 283 | 0.54 |
| ShortBetter | 67.94 | 333 | 58.90 | 250 | 71.70 | 260 | 65.86 | 376 | 46.92 | 405 | 14.36 | 387 | 54.28 | 335 | 0.41 |
| AdaThink-Med-Qwen | 68.34 | 79 | 58.74 | 109 | 73.50 | 62 | 66.45 | 121 | 48.72 | 175 | 14.16 | 88 | 54.99 | 106 | 0.93 |
| **AdaThink-Med-Qwen(+Llama)** | **69.12** | **61** | **59.87** | **95** | **73.50** | **35** | **66.88** | **89** | **48.58** | **143** | **14.33** | **75** | **55.38** | **83** | **1.00** |

*Table 17.* Experimental results for the radiology report generation task. Completeness is measured by the extent to which the generated report covers all required aspects, and redundancy reflects the level of unnecessary components in the generated text.

| Method | Completeness ↑ | Redundancy ↓ |
|---|---|---|
| Kimi | 66.2 | 42.5 |
| CosFn | 34.5 | 54.1 |
| DAST | 46.7 | 50.7 |
| ShortBetter | 55.3 | 45.2 |
| AdaThink-Med | **78.4** | **22.5** |

## G.7. Scalability Analysis on Long-Chain Reasoning Models

A critical consideration in efficient reasoning is the domain-specific nature of token consumption. In mathematical reasoning benchmarks (e.g., MATH500), high token counts are often driven by the verbose tokenization of numerical symbols and formulas. In contrast, medical reasoning relies primarily on plain text for differential diagnosis and evidence-based analysis, which is naturally more token-efficient. However, complex clinical queries still demand extended reasoning chains comparable to those found in mathematical tasks. Consequently, an effective compression strategy must be adaptive, capable of handling the high variance between simple factoid questions and intricate diagnostic scenarios.

To validate the scalability of AdaThink-Med on models that inherently generate extended reasoning chains, we conducted additional evaluations using HuatuoGPT-o1-8B and m1-7B. Unlike the standard instruction-tuned models used in the main experiments, these models are specifically optimized for long-context reasoning and exhibit significantly higher average output lengths.

The results, presented in Table 18, demonstrate the effectiveness of our approach in this high-token regime. AdaThink-Med successfully compresses the reasoning paths of the m1-7B model from an average of 2451 tokens to 1100 tokens, achieving a reduction of approximately 55% while maintaining an average accuracy of 58.03%. Similarly, for HuatuoGPT-o1-8B, our method reduces the average length from 527 to 203 tokens with minimal impact on performance. The consistently higher Accuracy-Efficiency Scores (AES) compared to baselines such as Kimi, CosFn, and ShortBetter confirm that our uncertainty-guided length calibration mechanism is robust and scalable, effectively optimizing models with diverse intrinsic reasoning length distributions.

*Table 18.* Performance comparison on long-chain reasoning models (HuatuoGPT-o1-8B and m1-7B). AdaThink-Med achieves significant length compression while maintaining competitive accuracy across six medical benchmarks. The best AES scores are highlighted in **bold**.

| Model | MedQA | | MedMCQA | | PubMedQA | | MMLU-ProM | | GPQA-M | | MedXpert | | Avg | | AES |
|---|---|---|---|---|---|---|---|---|---|---|---|---|---|---|---|
| | Acc. | Len. | Acc. | Len. | Acc. | Len. | Acc. | Len. | Acc. | Len. | Acc. | Len. | Acc. | Len. | |
| *HuatuoGPT-o1-8B Backbone* | | | | | | | | | | | | | | | |
| HuatuoGPT-o1-8B [CoT] | 76.35 | 568 | 62.82 | 445 | 79.80 | 445 | 63.71 | 521 | 54.35 | 585 | 17.06 | 601 | 59.02 | 527 | - |
| Kimi | 71.01 | 373 | 62.87 | 302 | 77.40 | 288 | 67.04 | 340 | 50.26 | 401 | 14.49 | 363 | 57.18 | 345 | 0.19 |
| CosFn | 68.42 | 260 | 62.63 | 126 | 77.30 | 159 | 65.60 | 220 | 48.97 | 187 | 14.24 | 246 | 56.20 | 200 | 0.38 |
| DAST | 71.17 | 730 | 61.01 | 619 | 76.70 | 578 | 64.36 | 697 | 45.90 | 794 | 15.02 | 765 | 55.69 | 697 | -0.60 |
| ShortBetter | 71.64 | 336 | 63.18 | 271 | 78.40 | 258 | 66.12 | 299 | 49.74 | 353 | 14.45 | 321 | 57.26 | 306 | 0.27 |
| AdaThink-Med | 72.74 | 183 | 63.54 | 192 | 76.60 | 72 | 66.91 | 204 | 50.77 | 344 | 13.88 | 221 | 57.41 | 203 | **0.48** |
| *m1-7B Backbone* | | | | | | | | | | | | | | | |
| m1-7B [CoT] | 75.01 | 2161 | 62.32 | 1749 | 74.20 | 1080 | 68.07 | 2564 | 51.53 | 3887 | 18.28 | 3265 | 58.24 | 2451 | - |
| Kimi | 71.23 | 1425 | 61.26 | 992 | 72.50 | 856 | 65.56 | 1335 | 50.58 | 2069 | 15.53 | 1974 | 56.11 | 1442 | 0.23 |
| CosFn | 72.58 | 1566 | 60.57 | 1011 | 72.70 | 774 | 67.99 | 1218 | 51.35 | 1655 | 16.23 | 1563 | 56.90 | 1298 | 0.36 |
| DAST | 70.87 | 2445 | 58.43 | 1995 | 70.10 | 1243 | 65.33 | 2655 | 48.33 | 4096 | 14.43 | 4096 | 54.58 | 2755 | -0.44 |
| ShortBetter | 73.55 | 1359 | 60.58 | 927 | 71.40 | 839 | 65.37 | 1290 | 50.12 | 1847 | 14.99 | 1885 | 56.00 | 1358 | 0.25 |
| AdaThink-Med | 74.15 | 982 | 63.35 | 876 | 73.10 | 585 | 67.99 | 1127 | 52.03 | 1596 | 17.55 | 1433 | 58.03 | 1100 | **0.53** |

## G.8. Benchmarking Against Closed-Source State-of-the-Art Models

To situate the performance and efficiency of AdaThink-Med within the broader landscape of current large language models, we extended our evaluation to include GPT-5 and Gemini-2.5-Pro. It is important to acknowledge that these proprietary models are trained on significantly larger datasets and possess parameter counts orders of magnitude higher than the 7B/8B scale models used in our main experiments. For the purpose of the Accuracy-Efficiency Score (AES) calculation in this subsection, we utilized the performance of the Qwen2.5-Instruct-7B model (fine-tuned with GRPO) as the baseline reference.

The comparative results are detailed in Table 19. We observe that while both closed-source models achieve high accuracy, their reasoning behaviors differ markedly. Gemini-2.5-Pro, despite its strong performance, tends to generate lengthy and often redundant reasoning chains, resulting in suboptimal inference efficiency. In contrast, GPT-5, which reportedly incorporates proprietary adaptive reasoning optimizations, demonstrates a significant reduction in output length while maintaining high accuracy.

Notably, AdaThink-Med (Llama backbone) achieves an inference efficiency profile comparable to that of GPT-5, reducing the average response length to just 64 tokens. While there remains a performance gap in absolute accuracy attributable to the vast disparity in model size (8B vs. hundreds of billions of parameters), the high AES scores obtained by our approach highlight its effectiveness. This demonstrates that AdaThink-Med successfully balances performance and computational cost, achieving state-of-the-art efficiency levels on smaller, open-weights models that are accessible for widespread deployment.

*Table 19.* Performance and efficiency comparison with closed-source state-of-the-art models (GPT-5 and Gemini-2.5-Pro). The Qwen-GRPO model serves as the baseline for AES calculations. AdaThink-Med achieves competitive efficiency scores despite the significant difference in model scale. Best AES scores are highlighted in **bold**.

| Model | MedQA | | MedMCQA | | PubMedQA | | MMLU-ProM | | GPQA-M | | MedXpert | | Avg | | AES |
|---|---|---|---|---|---|---|---|---|---|---|---|---|---|---|---|
| | Acc. | Len. | Acc. | Len. | Acc. | Len. | Acc. | Len. | Acc. | Len. | Acc. | Len. | Acc. | Len. | |
| Qwen-GRPO | 67.55 | 504 | 59.52 | 378 | 72.30 | 396 | 66.12 | 535 | 48.46 | 575 | 14.48 | 592 | 54.74 | 497 | - |
| GPT-5 | 76.07 | 239 | 62.89 | 198 | 68.00 | 278 | 72.42 | 316 | 59.13 | 320 | 35.25 | 335 | 62.46 | 281 | **+0.86** |
| Gemini-2.5-Pro | 74.20 | 505 | 62.60 | 408 | 70.80 | 220 | 68.12 | 549 | 57.32 | 549 | 27.91 | 612 | 60.16 | 474 | +0.34 |
| AdaThink-Med [Llama] | 67.00 | 36 | 60.41 | 73 | 78.40 | 54 | 63.58 | 85 | 47.94 | 100 | 16.20 | 35 | 55.59 | 64 | **+0.92** |
| AdaThink-Med [Qwen] | 68.34 | 79 | 58.74 | 109 | 73.50 | 62 | 66.45 | 121 | 48.72 | 175 | 14.16 | 88 | 54.99 | 106 | +0.80 |

## G.9. Ablation Study on Batch Normalization and Batch Size

To validate the architectural choices in AdaThink-Med, specifically the use of Batch Normalization (BN) and the impact of batch size, we conducted a series of ablation experiments. In our framework, BN serves two critical objectives: first, it constrains the output uncertainty to a normalized $[0, 1]$ interval, ensuring numerical stability; second, it works in conjunction with the Exponential Moving Average (EMA) to robustly estimate the global difficulty threshold of the dataset. This global

threshold is pivotal as it dictates the aggressiveness of the adaptive compression—datasets identified as globally "simpler" allow for greater length reduction.

Table 20 presents the comparative results of modifying batch normalization and batch size settings. Our analysis yields three key observations:

1. **Necessity of Batch Normalization:** The removal of BN (denoted as *AdaThink-Med-ablations(BN)*) leads to training instability. Without the normalization to a fixed range and the relative comparison provided by batch statistics, the algorithm fails to establish an accurate difficulty threshold. Consequently, the adaptive length penalty mechanism becomes ineffective, and the model's performance and output length revert to levels similar to the initial unoptimized model (AES +0.17), failing to achieve meaningful compression.

2. **Impact of Batch Size on Convergence:** Comparing the standard configuration (Batch Size $256 \times 8$) with a reduced setting ($32 \times 8$), we observe that batch size primarily influences the convergence velocity rather than the final performance. Larger batch sizes provide more representative statistics for the difficulty estimator, accelerating convergence (200 steps). However, smaller batch sizes, while slower to converge (280 steps), eventually achieve comparable efficiency and accuracy (AES +0.88 vs +0.92).

3. **Inefficacy of Per-Sample Calibration:** We also explored per-sample entropy calibration, which is theoretically equivalent to a batch size of 1. This approach resulted in high variance in difficulty estimation, leading to severe training instability and failure to converge. This confirms that the synergy between batch-level normalization and EMA is essential for robust global difficulty estimation and stable optimization.

*Table 20.* Ablation study on Batch Normalization (BN) and Batch Size. "Conv. Steps" denotes the number of steps required for convergence. Removing BN leads to optimization failure, while smaller batch sizes result in slower convergence but comparable final efficiency.

| Model | BN | Batch Size | MedQA Acc. | MedQA Len. | MedMCQA Acc. | MedMCQA Len. | PubMedQA Acc. | PubMedQA Len. | MMLU-ProM Acc. | MMLU-ProM Len. | GPQA-M Acc. | GPQA-M Len. | MedXpert Acc. | MedXpert Len. | Avg Acc. | Avg Len. | Conv. Steps | AES |
|---|---|---|---|---|---|---|---|---|---|---|---|---|---|---|---|---|---|---|
| Llama-3.1-8B-Instruct [CoT] | - | - | 68.18 | 461 | 57.32 | 309 | 78.20 | 296 | 60.06 | 457 | 45.64 | 485 | 16.08 | 545 | 54.25 | 425 | - | - |
| Llama-3.1-8B-Instruct | - | - | 54.75 | 302 | 54.69 | 169 | 77.00 | 113 | 57.00 | 256 | 35.64 | 345 | 14.48 | 342 | 48.93 | 254 | - | -0.09 |
| Llama-3.1-8B-Instruct [GRPO] | - | - | 72.50 | 438 | 61.60 | 299 | 78.50 | 290 | 63.51 | 442 | 46.92 | 461 | 17.26 | 529 | 56.72 | 410 | - | +0.17 |
| AdaThink-Med-ablations (BN) | × | $256 \times 8$ | 70.43 | 425 | 59.33 | 267 | 78.20 | 275 | 62.55 | 429 | 46.49 | 423 | 17.00 | 500 | 55.76 | 387 | Fluct. | +0.17 |
| AdaThink-Med-ablations (BS) | ✓ | $32 \times 8$ | 66.98 | 40 | 60.21 | 82 | 78.90 | 60 | 62.65 | 93 | 46.53 | 96 | 15.76 | 42 | 55.04 | 69 | 280 | +0.88 |
| AdaThink-Med | ✓ | $256 \times 8$ | 67.00 | 36 | 60.41 | 73 | 78.40 | 54 | 63.58 | 85 | 47.94 | 100 | 16.20 | 35 | 55.59 | 64 | 200 | +0.92 |

### G.10. Validation of Dataset Selection Strategy against Random Sampling

To verify that the performance gains observed in our dataset selection experiments stem from the intrinsic quality of the selected instances rather than merely the reduction in dataset size, we conducted a controlled comparison against random sampling. For each dataset retention ratio $\delta \in \{20\%, 40\%, 60\%, 80\%\}$, we constructed a control subset by randomly selecting samples from the full training corpus.

To ensure a fair comparison, we standardized the computational budget across all experiments by fixing the number of training steps at 200. This approach allows the model to achieve convergence regardless of the subset size, isolating the impact of data quality.

The results, presented in Table 21, demonstrate that our proposed dataset selection method consistently outperforms random selection across all retention ratios. For example, at $\delta = 40\%$, our method achieves an average accuracy of 55.12%, surpassing the random baseline (54.20%) and even slightly exceeding the full dataset baseline (54.74%). These findings confirm that AdaThink-Med effectively identifies high-value training samples that contribute most significantly to reasoning capability, validating the efficacy of our selection strategy beyond simple data reduction.

## H. Qualitative Results

### H.1. Case Studies

Fig. 6 presents a straightforward evaluation of basic genetics knowledge. The baseline models, m1 and HuatuoGPT-o1, display evident overthinking through numerous redundant reasoning steps, consuming 766 and 451 tokens respectively.

Autozygosity mapping is used to map disorders that show which pattern of inheritance? Options:
A: Autosomal recessive B: Y-linked recessive C: Incomplete dominance D: Mitochondrial inheritance E: Autosomal dominant F: Codominant inheritance G: X-linked recessive H: Multifactorial inheritance I: X-linked dominant J: Y-linked dominant

#766 tokens (m1-7B)

think\nOkay, so the question is asking about autozygosity mapping and which pattern of inheritance it's used for. Let me think. Autozygosity mapping involves looking at siblings who are identical twins, right? Because they share 100% of their genes, which makes it easier to identify regions of the genome associated with certain diseases.\n\nHmm, I remember that autosomal dominant disorders usually require only one copy of the defective gene to cause the disease. But since identical twins have both copies, maybe that's not the case here. Wait, but autozygosity might be more about finding regions where the disease is present in both twins but not in others. So perhaps it's used for recessive conditions because both alleles need to be defective. \n\nWait, autosomal recessive disorders typically require two copies of the gene to be affected. So if two siblings have the disease, maybe they both inherited the same recessive allele from their parents. Autozygosity mapping could help identify the specific chromosome region responsible. Let me check if that's correct.\n\nAlternatively, X-linked recessive disorders affect males more often. But since males have only one X, maybe that's not related directly to autozygosity. Autozygosity is more about having two copies of a chromosome segment. So, maybe autosomal recessive is the answer here. Wait, another thought: autozygous twins would have both alleles identical, so any disorder present would need both alleles to be susceptible. That aligns with recessive traits. Therefore, the answer should be A: Autosomal recessive.\n\nBut wait, wait. I recall that autozygosity mapping is particularly useful for multifactorial traits, which are influenced by multiple genes and environmental factors. No, wait, but the options given include H: Multifactorial inheritance. Hmm. Wait, multifactorial is usually about multiple genes and environment. Autozygosity, being about identical twins, would have the same genetic background. But multifactorial could still be possible because even with identical genomes, maybe epigenetics or other factors come into play. However, autozygosity is more commonly linked to recessive disorders because they require both alleles. For example, diseases like cystic fibrosis, which is autosomal recessive, were mapped using this method. \n\nSo I think the correct answer is A. Autosomal recessive. Let me cross-verify. Another way: in autosomal dominant, you only need one copy. If both twins have the disease, but autosomal dominant would mean one parent has it, but twins being 100% similar doesn't fit as well. Whereas in recessive, both twins have the two copies from parents, so mapping those regions makes sense. \n\nYes, definitely A.\nanswer\nAutozygosity mapping relies on studying siblings who are identical twins, as they share 100% of their genetic material. This method is particularly effective for identifying genetic regions associated with disorders that require **both alleles** of a gene to be defective. Such conditions are characterized by **autosomal recessive inheritance**, where an individual must inherit two copies of a mutated allele (one from each parent) to manifest the disease. Since identical twins provide a genetically identical background, autozygosity mapping is especially powerful for detecting recessive traits. Examples include diseases like cystic fibrosis. \n\nWhile multifactorial inheritance (option H) involves multiple genes and environmental factors, autozygosity mapping is not primarily used for such polygenic traits. Similarly, dominant inheritance patterns (options E, G, J) do not require both alleles to be defective, making them less relevant to autozygosity-based studies. \n\n**Answer:** \\boxed{A} √

#451 tokens (HuatuoGPT-o1-8B)

## Thinking\n\nOkay, so let's think about how autozygosity mapping works. This technique helps us figure out where a gene causing a disorder might be located on the genome by looking at people who have the same genetic makeup for two copies of the gene, like siblings who are affected by the same condition. That's interesting because it means we're dealing with recessive genes, since both copies need to be affected for the condition to show up.\n\nNow, if we're talking about autosomal recessive disorders, these are conditions where you need two copies of the mutated gene to actually see the symptoms. So, if someone has two copies of this gene, they'll definitely have the disorder. That's why autozygosity mapping is really useful here because it can pinpoint the location of these recessive genes.\n\nBut wait, what about other types of inheritance? Like, what about X-linked or Y-linked? Those involve different chromosomes, right? X-linked and Y-linked disorders don't work the same way because they're linked to specific sex chromosomes. So, autozygosity mapping wouldn't really help there because it focuses on autosomes, not sex chromosomes.\n\nAlso, what about dominant disorders? Well, those are different too. With autosomal dominant disorders, just one copy of the mutated gene is enough to cause the condition. So, you wouldn't need two copies to see the effect. And incomplete dominance or codominance? Those involve more complex interactions between different versions of a gene, so they wouldn't fit the autozygosity model either.\n\nSo, yeah, when it comes down to it, autozygosity mapping is all about finding recessive genes on autosomes. It's perfect for tracking down those pesky autosomal recessive disorders. That makes sense, right? It's all about identifying recessive conditions in people who have identical genetic backgrounds for the gene in question. So, the answer is definitely autosomal recessive.\n\n## Final Response\n\nThe correct answer is A: Autosomal recessive. Autozygosity mapping is particularly effective for identifying the genetic location of recessive genes on autosomes, making it well-suited for mapping disorders that follow an autosomal recessive pattern of inheritance. √

#45 tokens (AdaThink-Qwen)

Autozygosity mapping is particularly useful for identifying genetic disorders that follow an autosomal recessive pattern of inheritance because it focuses on regions of homozygosity that are shared by affected individuals.
\\boxed{A}", √

*Figure 6.* Case 1. AdaThink-Med delivers correct answer while consumes far fewer tokens than m1-7B and HuatuoGPT-o1-8B.

*Table 21.* Comparison of AdaThink-Med's dataset selection strategy against random subset selection across various retention ratios ($\delta$). To ensure fairness, all experiments were conducted with a fixed compute budget of 200 training steps. Our method consistently outperforms random selection, indicating the superior quality of the selected data.

| $\delta$ | Method | MedQA Acc. | MedQA Len. | MedMCQA Acc. | MedMCQA Len. | PubMedQA Acc. | PubMedQA Len. | MMLU-ProM Acc. | MMLU-ProM Len. | GPQA-M Acc. | GPQA-M Len. | MedXpert Acc. | MedXpert Len. | Avg. Acc. | Avg. Len. |
|---|---|---|---|---|---|---|---|---|---|---|---|---|---|---|---|
| 0% | - | 63.86 | 490 | 56.68 | 359 | 73.10 | 386 | 62.28 | 523 | 46.41 | 562 | 12.28 | 585 | 52.44 | 484 |
| 20% | Random | 64.21 | 505 | 57.23 | 384 | 73.50 | 391 | 62.88 | 537 | 46.99 | 588 | 13.22 | 577 | 53.01 | 497 |
| | Ours | 64.81 | 497 | 58.50 | 373 | 74.40 | 400 | 64.82 | 526 | 48.21 | 565 | 14.00 | 585 | 54.12 | 491 |
| 40% | Random | 64.38 | 523 | 57.66 | 358 | 72.30 | 415 | 66.65 | 515 | 50.22 | 557 | 13.99 | 585 | 54.20 | 493 |
| | Ours | 65.44 | 512 | 58.81 | 367 | 74.30 | 396 | 66.51 | 522 | 51.28 | 564 | 14.37 | 590 | 55.12 | 492 |
| 60% | Random | 64.96 | 507 | 57.43 | 352 | 72.60 | 416 | 65.38 | 532 | 48.32 | 532 | 14.02 | 551 | 53.79 | 482 |
| | Ours | 68.50 | 488 | 58.69 | 356 | 73.10 | 381 | 66.12 | 513 | 47.44 | 535 | 13.96 | 569 | 54.64 | 474 |
| 80% | Random | 66.38 | 499 | 56.73 | 366 | 71.20 | 398 | 66.33 | 515 | 48.96 | 528 | 14.67 | 560 | 54.05 | 478 |
| | Ours | 68.42 | 504 | 58.38 | 370 | 72.30 | 388 | 67.49 | 527 | 50.00 | 576 | 14.73 | 579 | 55.22 | 491 |
| 100% | - | 67.55 | 504 | 59.52 | 378 | 72.30 | 396 | 66.12 | 535 | 48.46 | 575 | 14.48 | 592 | 54.74 | 497 |

*Table 22.* Dataset selection experimental results. $\delta$ represents the ratio of training datasets.

| $\delta$ | In-domain MedQA Acc. | In-domain MedQA Len. | In-domain MedMCQA Acc. | In-domain MedMCQA Len. | Out-domain PubMedQA Acc. | Out-domain PubMedQA Len. | Out-domain MMLU-ProM Acc. | Out-domain MMLU-ProM Len. | Out-domain GPQA-M Acc. | Out-domain GPQA-M Len. | Out-domain MedXpert Acc. | Out-domain MedXpert Len. | Avg. Acc. | Avg. Len. |
|---|---|---|---|---|---|---|---|---|---|---|---|---|---|---|
| 0% | 63.86 | 490 | 56.68 | 359 | 73.10 | 386 | 62.28 | 523 | 46.41 | 562 | 12.28 | 585 | 52.44 | 484 |
| 20% | 64.81 | 497 | 58.50 | 373 | 74.40 | 400 | 64.82 | 526 | 48.21 | 565 | 14.00 | 585 | 54.12 | 491 |
| 40% | 65.44 | 512 | 58.81 | 367 | 74.30 | 396 | 66.51 | 522 | 51.28 | 564 | 14.37 | 590 | 55.12 | 492 |
| 60% | 68.50 | 488 | 58.69 | 356 | 73.10 | 381 | 66.12 | 513 | 47.44 | 535 | 13.96 | 569 | 54.63 | 474 |
| 80% | 68.42 | 504 | 58.38 | 370 | 72.30 | 388 | 67.49 | 527 | 50.00 | 576 | 14.73 | 579 | 55.22 | 491 |
| 100% | 67.55 | 504 | 59.52 | 378 | 72.30 | 396 | 66.12 | 535 | 48.46 | 575 | 14.48 | 592 | 54.74 | 497 |

In contrast, AdaThink-Med achieves a correct solution with only 45 tokens, demonstrating conciseness and accuracy. Furthermore, the baselines frequently produce incorrect conclusions during analysis and rely on a "wait" mechanism to initiate unnecessary rethinking cycles.

### H.2. Failure Case Studies

Upon careful examination, we identified that for medium-difficulty tasks, the model struggled with balancing non-thinking and thinking, often relying on quick thinking that resulted in insufficient reasoning. In failure case (a), the model's output incorrectly suggests that global management might improve outcomes based on trends toward better disease-free survival, despite the lack of statistically significant differences in overall survival and disease-free survival between global management and separate management. This reflects a failure in properly interpreting the study results, where no significant differences in survival were found, and the disease-free survival advantage for SM was statistically significant. The model appears to overemphasize minor trends, neglecting the statistical evidence that showed global management did not provide superior results in the key outcomes. This oversight underscores the importance of distinguishing between statistical significance and potential clinical trends, particularly when the data does not support a definitive advantage for one management strategy over the other. Consequently, a more nuanced approach should have been employed to align the analysis with the study's findings. In failure case (b), the model incorrectly identifies the S3 gallop (E) as the most specific sign for pulmonary arterial hypertension. While an S3 sound can indicate left ventricular dysfunction, it is more commonly associated with volume overload or heart failure, not specifically PAH. In contrast, the sign of P2 louder than A2 (H) is far more specific to pulmonary arterial hypertension, as it directly reflects increased pressure in the pulmonary arteries. The model failed to consider this crucial auscultatory finding, which is the hallmark sign of PAH, particularly as the increased pulmonary pressure causes delayed and louder closure of the pulmonary valve (P2). Thus, the model did not appropriately prioritize the

A 51-year-old man presents the emergency room with chest pain. He mentions that the pain started several hours ago and radiates to his left neck and shoulder. He also mentions that he has some difficulty in breathing. He says that he has had similar chest pains before, but nothing seemed to be wrong at that time. He was diagnosed with high cholesterol during that episode and was prescribed medication. He also has a 3-year history of gastritis. The blood pressure is 130/80 mm Hg, respirations are 18/min, and the pulse is 110/min. He seems a little anxious. The physical examination reveals no significant abnormalities. An ECG shows slight changes in the leads. His physician talks to him about the benefits of taking low-dose aspirin daily. Which of the following would be a contraindication to the use of aspirin in this patient? Options: A: ECG changes B: Hypercholesterolemia C: Gastritis D: Increased pulse rate.

#346 tokens (UltraMedical-8B-3.1)

To address the question about contraindications for the use of low-dose aspirin in this patient, we need to consider each option in the context of known risks associated with aspirin therapy:\n\nA. **ECG changes**: Aspirin is commonly used in the setting of ischemic heart disease due to its antiplatelet effects, which can prevent clot formation. ECG changes could suggest underlying heart disease, such as angina or myocardial infarction, where aspirin might be beneficial rather than contraindicated.\n\nB. **Hypercholesterolemia**: High cholesterol itself is not a direct contraindication to aspirin therapy. However, the use of aspirin in patients with hypercholesterolemia is more related to cardiovascular risk reduction rather than a direct contraindication.\n\nC. **Gastritis**: Aspirin can exacerbate gastrointestinal issues because it inhibits prostaglandin synthesis, which protects the gastric lining. In patients with a history of gastritis, aspirin can increase the risk of gastrointestinal bleeding. This makes aspirin potentially harmful in the context of existing gastrointestinal conditions.\n\nD. **Increased pulse rate**: While aspirin can occasionally cause an increase in heart rate, this is not a contraindication for its use. The primary concern with increased pulse rate would be the potential for underlying cardiac issues, but this does not directly contraindicate aspirin use.\n\nGiven these considerations, the most relevant contraindication among the options provided is the patient's history of gastritis (C), as aspirin can exacerbate gastrointestinal bleeding, especially in someone who already has a compromised gastric lining.\n\n**So, the answer is C.** √

#496 tokens (HuatuoGPT-o1-8B)

## Thinking\n\nAlright, let's think about this. We've got a 51-year-old guy who's come into the ER with chest pain. He says it started a few hours ago and it's spreading to his left neck and shoulder. That sounds like classic heart issues to me. And he's having trouble breathing too. Not good.\n\nHe's had these chest pains before, but nothing showed up back then. They just told him he had high cholesterol and gave him meds for it. Okay, so maybe it's something new or getting worse. And he's been dealing with gastritis for three years. That's not great, but let's see how it plays into things.\n\nHis vitals are interesting. Blood pressure is okay at 130/80, but his heart rate is up at 110. That could mean he's anxious or there's something more going on. The doctor mentioned something about slight ECG changes. Hmm, that makes me think we're definitely looking at some kind of cardiac issue here.\n\nNow, the doctor wants to put him on low-dose aspirin. That's a common move when you're worried about a heart attack. But wait, what about the gastritis? Aspirin can really irritate the stomach lining, especially if someone already has gastritis. That's a big concern because it could make his condition worse.\n\nLet's think about the other stuff. High cholesterol is a risk factor for heart disease, so aspirin might actually help with that. And an increased pulse rate could be a sign of stress or anxiety, but it's not a dealbreaker for aspirin. So, those aren't reasons to say no to aspirin.\n\nThe ECG changes are a bit tricky. They suggest something's up with the heart, which is why they want to start aspirin. It's not a reason to avoid it; it's a reason to give it.\n\nSo, putting it all together, the big red flag here is the gastritis. Aspirin can cause stomach problems, and with his history of gastritis, it's not a good idea. Yeah, that's the one that stands out as a contraindication.\n\n## Final Response\n\nThe contraindication to the use of aspirin in this patient is C: Gastritis. Aspirin can exacerbate gastric issues, particularly in someone with a history of gastritis, making it a significant concern in this case. √

#49 tokens (AdaThink-Qwen)

The most common contraindication for aspirin use in patients with a history of gastrointestinal issues like gastritis is the risk of exacerbating these conditions due to aspirin's potential to cause gastric irritation or bleeding.\n\n\\boxed{C} √

*Figure 7.* Case 2. AdaThink-Med delivers correct answer while consumes far fewer tokens than UltraMedical-8B-3.1 and HuatuoGPT-o1-8B.

A 31-year-old man comes to the emergency department because of acute onset neck pain and enlargement. Specifically, he reports that he has been experiencing pain and swelling of the anterior portion of his neck near the midline. Otherwise, he says that he has been getting tired easily and feels cold often. Physical exam reveals a painful diffusely enlarged thyroid gland with many small nodules. A biopsy is obtained showing diffuse cellular hyperplasia with areas of focal colloid hyperplasia. Given these findings, the patient is started on appropriate therapy, and the neck mass becomes smaller over time. Which of the following is most likely associated with the cause of this patient's symptoms?\nOptions:\nA: HLA-B8 risk factor\nB: Iodine deficiency\nC: Presence of embryological remnant\nD: Proliferation of fibroinflammatory cells

#### #406 tokens (UltraMedical-8B-3.1)

The clinical presentation of acute onset neck pain and enlargement of the thyroid gland, along with symptoms of fatigue and feeling cold, suggests a thyroid dysfunction. The physical examination finding of a diffusely enlarged thyroid gland with many small nodules points towards a thyroiditis or a goiter rather than a solitary nodule.\n\nThe biopsy result showing diffuse cellular hyperplasia with areas of focal colloid hyperplasia further supports the diagnosis of a thyroiditis, specifically a form that involves the entire gland (diffuse). This can be consistent with Hashimoto's thyroiditis, which is an autoimmune condition and can present with these findings. However, the acute onset and painful nature of the enlargement are more suggestive of subacute thyroiditis (also known as de Quervain's thyroiditis), which is often associated with a viral infection.\n\nSubacute thyroiditis is not typically associated with HLA-B8 (which is more commonly associated with autoimmune conditions like type 1 diabetes and celiac disease) or the presence of an embryological remnant (which would suggest a congenital or developmental anomaly leading to a thyroglossal duct cyst or ectopic thyroid tissue).\n\nIodine deficiency is a common cause of goiter, but it typically does not cause a painful thyroid gland and would not explain the acute presentation. It also tends to cause a symmetrical enlargement of the thyroid rather than multiple nodules.\n\nProliferation of fibroinflammatory cells is characteristic of subacute thyroiditis, where there is a granulomatous inflammation due to the infiltration of the thyroid gland by inflammatory cells, including lymphocytes, plasma cells, and macrophages.\n\nGiven the clinical presentation and biopsy findings, the most likely cause of the patient's symptoms is subacute thyroiditis. The patient's symptoms and the response to treatment (with the neck mass becoming smaller over time) support this diagnosis. Subacute thyroiditis often resolves on its own, although symptomatic treatment may be necessary during the acute phase. \n\nSo, the answer is D. ✗

#### #436 tokens (HuatuoGPT-o1-8B)

## Thinking\n\nAlright, so we have a 31-year-old guy who's suddenly got neck pain and swelling right in the front, near the middle. He's also feeling really tired and cold a lot. That's interesting. When we check him out, his thyroid is all swollen and lumpy, like there are lots of little nodules. They did a biopsy, and it shows a bunch of cells growing too much and some areas with extra colloid. This sounds like a classic case of Graves' disease. \n\nGraves' disease is known for causing the thyroid to get all overactive and swell up. It's an autoimmune thing where the body's immune system goes a bit haywire and starts attacking the thyroid. So, it makes sense that he'd feel tired and cold because his metabolism is probably going into overdrive. \n\nNow, let's think about what might be causing this. Graves' disease is often linked to certain genetic factors. People with HLA-B8 are more prone to developing it. That's a specific genetic marker that increases the risk. \n\nBut wait, there's also iodine deficiency to consider. It can affect the thyroid, but it usually causes a different kind of problem, more like a goiter or hypothyroidism, not hyperthyroidism. So, that doesn't seem to fit here. \n\nWhat about embryological remnants? Those are more about things like thyroglossal duct cysts, which don't really match our guy's situation. And fibroinflammatory cells? That's more about inflammation, not the kind of cellular hyperplasia we're seeing here. \n\nSo, putting it all together, the most likely cause of his symptoms is the HLA-B8 risk factor. It's a genetic predisposition that fits with the autoimmune nature of Graves' disease. Yeah, that makes sense.\n\n## Final Response\n\nThe most likely association with the cause of this patient's symptoms is **A: HLA-B8 risk factor**. The symptoms and biopsy findings suggest Graves' disease, an autoimmune condition that often has a genetic component, with HLA-B8 being a known risk factor. ✗

#### #156 tokens (AdaThink-Qwen)

Based on the clinical presentation, physical exam, and biopsy findings, the most likely diagnosis is Hashimoto's thyroiditis, which can present with a painful, diffusely enlarged thyroid with nodularity and colloid hyperplasia.\n\nThe correct association is:\n\nB: Iodine deficiency\n\nHowever, iodine deficiency is more commonly associated with goiter formation without the specific symptoms described, so it is less likely given the specific symptoms presented. The most probable cause is an autoimmune disorder leading to Hashimoto's thyroiditis.\n\nGiven options and the context, the closest association for the cause would be related to immune-mediated disease.\n\nFinal answer: \\boxed{B} (though iodine deficiency directly isn't the most common direct trigger, B is the closest among the options provided). ✓

*Figure 8.* Case 3. An overthinking case of UltraMedical-8B-3.1 and HuatuoGPT-o1-8B.

Intestinal lactase deficiency leads to lactose maldigestion which subsequently can lead to lactose intolerance, i.e. causing clinical symptoms. Infants and children normally express sufficient lactase levels as not to experience clinical symptoms. Overall, adults can be classified as lactase-persisters or lactase non-persisters. What is the approximate percentage of Lactase non-persisters worldwide?\n\nOptions:\nA: 30%-35%\nB: 80%-85%\nC: 90%-95%\nD: 55%-60%\nE: 5%-10%\nF: 45%-50%\nG: 60%-65%\nH: 10%-15%\nI: 20%-25%\nJ: 70%-75 %\n

### #1926 tokens (m1-7B)

think\nOkay, let's see. The question is asking about the approximate percentage of lactase non-persisters worldwide. Hmm. I remember that lactase non-persister is the condition where people lose the ability to produce lactase after infancy, leading to lactose intolerance. So, lactase persisters are those who can digest lactose into adulthood.\n\nFrom what I've studied before, this varies a lot depending on ethnicity and geography. For instance, in populations with a history of dairying, like Northern European descent, lactase persistence is more common. Conversely, in many Asian, African, and South American populations, lactase non-persistence is more prevalent because there wasn't as much selective pressure to retain that ability.\n\nThe options given are percentages between 5% to 95%, so it's about which broad range is correct. Let me recall some numbers. I think that around 60-70% of the world's population becomes lactase non-persistent after infancy. That would put it in the range of option J: 60-65%, but wait, option J says 70-75%. Wait, maybe I should check again.\n\nWait, another angle: in many studies, it's often cited that about 70% of the global population has lactase non-persistance. For example, in regions like East Asia, it's almost everyone, while in Europe, it's around 20%. Merged globally, I think the average is roughly 70% or higher. But sometimes different sources might say slightly different numbers.\n\n⋯⋯way. The World Health Organization (WHO) might have data on this, but I don't have access right now. From my prior education, I remember that the majority of adults globally are lactase non-persisters, with estimates around 70%. Therefore, I think option J:70-75% is correct. But to confirm, perhaps cross-checking with some percentages.\n\nFor example, in a study from the National Academy of Sciences, it states that approximately 75% of the world's population lacks the ability to fully digest lactose after infancy. So that would support option J. However, if the exact number is 70%, then J includes that. Therefore, the answer is J. However, if another source says 65%, then G. But without precise data, relying on commonly cited figures. Since many sources point to around 70-75%, I will go with J.\n\nanswer\n\nAnswer: The percentage of lactase non-persisters worldwide refers to individuals who lose lactase production after infancy, leading to lactose intolerance. This condition varies significantly by population, with higher rates in regions lacking a historical context of dairy consumption (e.g., Asia, Africa) and lower rates in lactase-persister populations (e.g., Northern Europe).\n\nGlobal estimates average around **70–75%** of adults being lactase non-persisters, supported by widespread references in human biology and medical literature. While sub-regional variations exist (e.g., >90% in East Asia, ~20% in Northern Europe), the overall global average aligns closely with **70–75%**, making **Option J** the correct choice.\n\n\\boxed{J}  ✘

### #453 tokens (HuatuoGPT-o1-8B)

## Thinking\n\nOkay, let's think about this. So, we're looking at lactase deficiency and how it relates to lactose intolerance. Lactase is that enzyme that helps break down lactose in milk, right? If you don't have enough of it, you might feel pretty uncomfortable after eating dairy.\n\nNow, infants and young kids usually have plenty of lactase, so they don't really have issues with milk. But as people grow up, some of them start to lose this enzyme. That's when we get these two groups: lactase-persisters and lactase non-persisters. The persisters keep making lactase into adulthood, while the non-persisters don't.\n\nSo, what's the deal with the numbers? I've heard that a lot of people around the world are lactase non-persisters. It's a pretty common thing, especially in certain parts of the world like Africa, Asia, and parts of Europe. But how many exactly?\n\nHmm, I remember reading that somewhere between 70% and 80% of the world's population falls into this category. That seems like a big chunk of people. But wait, I should double-check that number. It's always good to verify.\n\nLet me think. Some sources say it's even higher, like 90% in some places. But I'm not sure if that's right. I need to dig deeper.\n\nOh, here's something interesting. In some regions like Europe and North America, it's much lower, like around 5-10%. That's because those populations have more dairy in their diets, so there's more selective pressure to keep the lactase gene active.\n\nSo, if I put all this together, it seems like globally, the percentage of lactase non-persisters is somewhere between 70% and 85%. Yeah, that feels right. It's a pretty high number, but it makes sense given the global distribution of lactase activity.\n\nAlright, I'm confident with that answer. The approximate percentage of lactase non-persisters worldwide is likely around 80-85%.\n\n## Final Response\n\nThe approximate percentage of lactase non-persisters worldwide is B: 80%-85%.  ✓

### #116 tokens (AdaThink-Qwen)

To determine the approximate percentage of lactase non-persisters worldwide, we need to consider the global distribution of this trait. It's well-established that lactase persistence is more common in populations with a history of cattle domestication, such as Northern Europe, while lactase non-persistence is more common in other regions.\n\nHowever, for a general estimate without specific regional focus, the most widely cited global prevalence of lactase non-persistence is around 70-80%. This aligns best with option J (70%-75%).\n\n\\boxed{J}  ✓

*Figure 9.* Case 4. An overthinking case of HuatuoGPT-o1-8B.

**Question:** Synchronous liver metastases (SLM) occur in 20% of colorectal cancers (CRC). Resection of SLM and CLC can be undertaken at different centres (separate management, SM) or at the same centre (global management, GM). Retrospective study of SLM and CRC resections carried out during 01/2000 - 12/2006 by SM or GM, using a combined or delayed strategy. Morphologic characteristics and type of CRC and SLM resection were similar for the GM (n = 45) or SM (n = 66) groups. In patients with delayed liver resection (62 SM, 17 GM), chemotherapy prior to liver surgery was used in 92% and 38% of SM and GM patients (P<0.0001) and the median delay between procedures was 212 and 182 days, respectively (P = 0.04). First step of liver resection was more often performed during colorectal surgery in the GM group (62 vs. 6% for SM, P<0.0001) and the mean number of procedures (CRC+SLM) was lower (1.6 vs. 2.3, P = 0.003). Three-month mortality was 3% for GM and 0% for SM (n.s.). Overall survival rates were 67% and 51% for SM and GM at 3 years (n.s.), and 35 and 31% at 5 years (n.s.). Disease-free survival to 5 years was higher in SM patients (14% vs. 11%, P = 0.009).Using the above document as a reference, please review the following question and choose the option you find most appropriate. Colorectal cancer with synchronous liver metastases: does global management at the same centre improve results? Options: A: yes B: no C: maybe

**Ground Truth: B**

**Response:** Based on the information provided, the overall survival and disease-free survival rates did not significantly differ between the global management and separate management groups. However, the number of procedures was lower and chemotherapy use before liver surgery was more frequent in the global management group, which could imply better coordination and possibly better outcomes. Given this context, the best answer is:\\boxed{A}While the survival rates are not significantly different, the trend towards better disease-free survival suggests that global management might have advantages, even if not statistically significant for all measures. ✗

(a) Failure Case 1

**Question:** A 37-year-old woman with right lower extremity edema is evaluated because of the sudden onset of shortness of breath and pleuritic chest pain. A diagnosis of pulmonary embolism is made. Which of the following signs, if present on physical examination, would be the most specific indicator of pulmonary arterial hypertension in this patient? Options: A: Blue lips or skin B: Peripheral edema C: Rapid heart rate D: Constant chest pain E: Presence of an S3F: Wheezing or whistling sound when breathing G: Shortness of breath after slight activity H: P2 louder than A2I: Increased jugular venous pressure J: Coughing up blood. Please reason step by step, and put the final answer in \\boxed{}.

**Ground Truth: H**

**Response:** The most specific sign for pulmonary arterial hypertension in this scenario would be an S3 gallop heard on auscultation, as it suggests left ventricular dysfunction which can occur secondary to chronic pulmonary hypertension.\\boxed{E}. ✗

(b) Failure Case 2

*Figure 10.* Failure Cases Study

most specific sign for PAH.

