# OpenReview forum: "Optimizing Inference-Time Compute for Medical Reasoning via Uncertainty Quantification"
_ICML.cc/2026/Conference — ICML 2026 regular_

### Official Review · Reviewer_ymrL · 2026-03-12

**Soundness:** 3
**Presentation:** 3
**Significance:** 2
**Originality:** 2
**Overall Recommendation:** 4
**Confidence:** 3

**Summary:**

This paper presents AdaThink-Med, an uncertainty-guided RFT framework for reducing inference-time compute in medical reasoning models. The method estimates problem difficulty by combining rollout correctness with token-level entropy, and then calibrates the reward to encourage short responses for confident and simple cases while preserving longer reasoning for uncertain or difficult ones. Experiments on six medical QA benchmarks across Llama and Qwen backbones show substantial reductions in response length with competitive  accuracy.

**Compliance With Llm Reviewing Policy:**

Affirmed.

**Final Justification:**

The author's rebuttal addressed my initial concerns.

**Key Questions For Authors:**

I have listed all my concerns in previous sections and wish the authors to address.

**Limitations:**

Yes

**Strengths And Weaknesses:**

- Strengths

1. The paper addresses a practical and important problem: improving the inference efficiency of LLMs in the medical domain. Specifically, medical reasoning models often over-generate reasoning tokens, incurring unnecessary inference cost.

2. The proposed method uses the entropy (uncertainty) of reasoning trajectories together with accuracy to gauge the appropriate thinking length and apply corresponding penalties. Using uncertainty in conjunction with correctness as a joint signal is a sensible and well-motivated design choice.

3. The empirical results are comprehensive. Experiments across two backbones and six benchmarks provide solid evidence that the method improves reasoning efficiency in the medical domain.

- Weaknesses

1. A significant concern is that the control mechanism for efficiency feels somewhat forced. The core of the method involves pre-defining problem difficulty and applying manual penalties based on that difficulty (i.e., forcing shorter reasoning for "easy" questions). In my view, a more "natural" approach would involve designing a reward signal that encourages the model to autonomously explore the most accurate and token-efficient paths during training.

2. The difficulty estimator relies on rollout correctness and entropy statistics, which increases algorithmic complexity. Furthermore, entropy can be influenced by many factors, such as response length or specific reasoning trajectories. High entropy does not strictly imply low confidence in the final answer, and a model might have high path uncertainty but still consistently arrive at the correct conclusion. Perhaps the authors could consider using answer consistency across rollouts as a more robust measure of uncertainty?

3. The authors should provide more granular details regarding the RL training process, including the specific framework used and hyperparameter settings. Additionally, the paper would benefit from stronger controlled baselines where the standard RL adaptation budget is applied. I would also like to see a comparison with other length-control rewards that utilize uncertainty, what specifically differentiates this approach from existing ones?

---

> ### Author Rebuttal · Authors · 2026-03-31
>
> We sincerely thank Reviewer ymrL for the feedback. We would like to respectfully clarify several misunderstandings regarding our method below.
>
> **W1:** We explain why the difficulty threshold is necessary from three angles:
>
> **(1) Model capability varies across questions.** The same question may be trivial for a strong model but challenging for a weaker one. A uniform brevity reward penalizes all long outputs equally, including those where extended reasoning is needed. Our threshold, dynamically computed from the model's own uncertainty and correctness (Eq. 3), reflects *current* capability on each question, enabling targeted calibration.
>
> **(2) Tasks have intrinsically different difficulty.** Medical queries range from simple retrieval to complex differential diagnoses. A threshold-free reward treats all questions identically, over-compressing hard cases or under-compressing easy ones. Our $\tau$-quantile threshold adapts to the difficulty distribution within each training batch.
>
> **(3) Without a threshold, efficiency is uncontrollable.** As shown in Section 4.1 and Figure 2, unconstrained efficiency rewards (Kimi1.5, ShortBetter) cause catastrophic reward hacking—models collapse into meaningless short strings, destroying reasoning entirely. The threshold prevents this by protecting necessary reasoning on hard questions while compressing easy ones.
>
> Importantly, this threshold is **only used during training**. At inference, the model autonomously decides reasoning depth—as shown in Fig. 3(a), it naturally diverges into "thinking" and "non-thinking" modes without any external routing, achieving the self-guided efficiency the reviewer envisions.
>
> **W2:** Our difficulty estimator (Eq. 3) already combines entropy **and** correctness. We chose entropy as the additional signal beyond correctness for specific reasons:
>
> 1. **Negligible Overhead:** Entropy is practically "free"—logits are already computed during RL forward passes; entropy requires only basic tensor operations, adding <1% overhead. No additional forward passes needed.
> 2. **Path Uncertainty is a Feature:** A model exhibiting high entropy but correct answers represents an *internally difficult* question requiring careful reasoning. If measured solely by answer consistency, this would be misclassified as "easy," triggering length penalty and premature truncation—precisely the reward hacking shown in Figure 2. Capturing path uncertainty protects the model's reasoning on challenging problems.
> 3. **Granularity:** With $G$=8 rollouts, answer consistency yields discrete signals (12.5% increments). Token-level Top-K entropy provides a smooth, continuous gradient crucial for stable optimization. Furthermore, extracting/matching final answers across diverse trajectories introduces parsing noise, while entropy leverages the mathematically rigorous probability distribution directly, requiring no answer parsing.
>
> **W3:** (1) **Training Details:** All configurations and hyperparameters are **already documented in Appendix E.1 and E.2** with supplementary code. We will add a brief "Implementation Summary" in the main text for readability.
>
> (2) **Controlled Baselines:** Extending standard GRPO from convergence (300 steps) to 500 steps—matching AdaThink-Med's total compute—yields negligible improvement:
>
> |Backbone|GRPO(300) Acc/Len|GRPO(500) Acc/Len|AdaThink-Med Acc/Len|
> |:---|:---|:---|:---|
> |Llama-3.1-8B|56.72%/410|56.73%/408|**55.59%/64**|
> |Qwen-2.5-7B|54.74%/497|54.71%/494|**54.99%/106**|
>
> GRPO(500) barely changes accuracy or length, confirming our gains stem from the adaptive length reward design, not additional training compute.
>
> (3) **Differentiation from existing entropy-based methods:**
> - **vs. SEED-GRPO:** SEED-GRPO uses semantic entropy to modulate the **magnitude of policy updates** (e.g., smaller gradients for uncertain queries), without altering the reward itself. AdaThink-Med fuses Top-K token entropy with correctness to redesign the **RL reward function**, actively teaching the model to adjust generation length based on difficulty.
> - **vs. Adaptive Termination:** This is strictly a **test-time intervention** that externally halts parallel reasoning branches. AdaThink-Med is a **training-time solution**—we internalize length planning into the model's weights via RL, enabling autonomous "thinking/non-thinking" mode switching in a single pass without external controllers.
> - **vs. Step Entropy Compression:** This approach inserts `[SKIP]` tokens to prune low-entropy steps, but rigid pruning breaks reasoning on hard problems. AdaThink-Med operates at the sequence-level, fusing uncertainty with correctness into a Difficulty Score that provides **"performance compensation"**—compressing easy queries while incentivizing extended reasoning on complex cases, avoiding accuracy collapse from rigid pruning.
>
> We will expand Related Works to detail these distinctions. We sincerely appreciate the reviewer's evaluation and welcome any further discussion.

---

> > ### Author Rebuttal · Reviewer_ymrL · 2026-04-03
> >
> > Thank you for your reply. I have raised my score.

---

> > > ### Author Response · Authors · 2026-04-03
> > >
> > > Thank you for your review and the the accept recommendation. We are pleased that our rebuttal has addressed your concerns.

---

### Official Review · Reviewer_6MLs · 2026-03-13

**Soundness:** 3
**Presentation:** 3
**Significance:** 3
**Originality:** 3
**Overall Recommendation:** 5
**Confidence:** 4

**Summary:**

In this work, the authors propose AdaThink-Med, an end-to-end framework that enables medical LLMs to adaptively calibrate their reasoning length based on query complexity. Using entropy-based uncertainty estimation within reinforcement fine-tuning, the method penalizes verbose reasoning for straightforward cases while encouraging deeper exploration for ambiguous ones. The approach significantly reduces inference token consumption across multiple benchmarks with minimal performance loss, and the model spontaneously learns to switch between concise and extended reasoning modes based on clinical urgency.

**Compliance With Llm Reviewing Policy:**

Affirmed.

**Final Justification:**

The author's rebuttal has addressed all my concerns, and I've raised the score from 4 to 5.

**Key Questions For Authors:**

Please see weaknesses. My main concerns are 1. the use of uncertainty as a measure of difficulty is insufficiently motivated. 2. the results in the main table seems to show a performance reduction in a few datasets. I would consider raising my score if both of these issues are addressed.

**Limitations:**

The author should include a more detailed impact statement, as this method has potential usability in the healthcare domain.

**Strengths And Weaknesses:**

Strength

- The author did a comprehensive discussion of prior works in Sec. 2, comparing the current method with other methods on entropy guided reasoning.

- The use of entropy as a proxy for uncertainty/difficulty is a novel idea that has not been well explored before.

- The experimental datasets are comprehensive and covers most of the popular datasets.

Weakness

- The authors seem to use uncertainty as a proxy of difficulty. The two measures may not be always positively correlated, especially when the uncertainty is aleatoric. For example, an blurred/noisy X-ray image may make the model fairly uncertain, but this does not mean the model should spend more reasoning budget on it. On the other hand, it’s also possible to use the dynamic (rather than static) mean reward as a direct measure of difficulty (e.g. https://arxiv.org/abs/2506.00711). Can the authors clarify the motivation behind using uncertainty instead of using mean reward?

- While AdaThink shows impressive length reduction, it seems to come at a cost of lower accuracy, as shown in Table 1 and 2. In critical domain like medical analysis, it’s hard to justify the use of the proposed method, if results show it will reduce the accuracy of the model.

- Figure 2 is a bit hard to understand when length & accuracy are mixed together in a single figure. It’d be better if it can be separated into multiple ones.

- The author should include a more detailed impact statement, as this method has potential usability in the healthcare domain.

---

> ### Author Rebuttal · Authors · 2026-03-31
>
> We sincerely thank Reviewer 6MLs for the constructive feedback.
>
> **W1:** We respectfully note that this concern may stem from a misunderstanding of our method. We **do not** use uncertainty as the sole proxy for difficulty. As detailed in **Eq. 3**, our difficulty metric explicitly combines **dynamic uncertainty** with **dynamic correctness** (mathematically equivalent to the dynamic mean reward suggested). Both are calculated dynamically online from the model's own real-time rollouts, not from any static or pre-computed labels.
>
> The motivation for integrating uncertainty on top of mean reward is twofold:
> 1. **Granularity in Small Rollouts:** With typical group size $G$=8, mean reward yields discrete, coarse signals (increments of 12.5%). Token-level entropy provides a smooth, continuous gradient for difficulty estimation, which is crucial for stable policy optimization.
> 2. **Distinguishing "Lucky Guesses" from Mastery:** A model might guess correctly across rollouts (high mean reward) but exhibit high internal entropy. Factoring in uncertainty correctly identifies these fragile successes as "hard," preventing premature truncation of reasoning chains on unmastered concepts.
>
> We also note that the reference (https://arxiv.org/abs/2506.00711) suggested by the reviewer proposes using dynamic mean reward as a difficulty measure. Our Eq. 3 already incorporates this signal via the correctness component—the key distinction is that we further augment it with entropy to address the granularity and "lucky guess" issues above. For aleatoric uncertainty concerns (e.g., blurred X-rays), the correctness term naturally moderates the difficulty score, and the $\alpha$ parameter explicitly controls this balance—our ablation (Table 4) confirms $\alpha$=0.5 outperforms both correctness-only ($\alpha$=0.1, AES drops from 0.93 to 0.78) and entropy-only ($\alpha$=1.0) variants.
>
> **W2:** In adaptive reasoning, accuracy-efficiency trade-off is inherent—our method does not universally degrade performance but achieves the **optimal Pareto trade-off**. On Qwen2.5-7B, AdaThink-Med **improved** accuracy from 54.74% to 54.99% while achieving 4.7x token reduction. Llama shows only a 1.13% dip, offset by 6.4x acceleration—and notably, on the challenging GPQA-M benchmark, Llama accuracy actually increased from 46.92% to 47.94%. Crucially, as shown in Fig. 3(a), our model autonomously produces longer reasoning chains on complex tasks (e.g., GPQA-M: 100/175 tokens on Llama/Qwen) while compressing simple ones (e.g., MedQA: 36/79 tokens). This adaptive allocation ensures that for difficult medical analysis, the model retains sufficient reasoning depth. AdaThink-Med achieves the highest AES (+0.92/+0.93) across all baselines, with the next best (Kimi1.5) at only +0.78/+0.55—confirming superior Pareto efficiency.
>
> To directly address the concern about safety in medical analysis, our blind Human Evaluation by three board-certified clinicians (Table 4) confirms 89.00% Reasoning Sufficiency, far surpassing all baselines (Kimi: 73.00%, ShortBetter: 72.00%). We also conducted a **Logical Soundness (LS)** evaluation measuring whether reasoning chains are free from medical hallucinations (detailed in our Response to Reviewer 1, Q4): AdaThink-Med achieves 86.40% LS vs. <70% for aggressive baselines that exhibit "shortcut hallucinations"—fabricating concise but medically flawed rationales to exploit brevity rewards. This demonstrates that our uncertainty-guided mechanism preserves rigorous clinical logic: it only compresses reasoning when the model is both correct and confident, ensuring compression relies on internalized knowledge rather than spurious shortcuts.
>
> **W3:** We appreciate this constructive suggestion. In the revision, we will separate Figure 2 into dedicated subplots for length and accuracy to improve clarity.
>
> **W4:** We will replace the current brief impact statement with an expanded version covering: (1) how AdaThink-Med reduces latency and cost, democratizing access to reasoning models for resource-constrained clinical workflows (e.g., emergency triage, routine diagnostics); (2) environmental benefits from reduced redundant computation; (3) explicit emphasis that AdaThink-Med is a decision-support tool requiring human-in-the-loop oversight and rigorous clinical validation before deployment.
>
> **Q1:** We believe our responses to W1 (dual-metric design with ablation evidence) and W2 (Pareto-optimal trade-off with clinical safety validation) directly address both concerns. We hope these clarifications adequately address the reviewer's questions.
>
> We sincerely appreciate the reviewer's time and thoughtful evaluation. The questions raised have helped us identify areas where our presentation can be significantly improved. We are committed to incorporating all the discussed revisions into the final version and welcome any further suggestions during the discussion period.

---

> > ### Author Rebuttal · Reviewer_6MLs · 2026-03-31
> >
> > Thanks authors for the response. Most of my concerns have been addressed. I have a few minor followup questions:
> >
> > 1.  The author mentioned
> >
> > > our ablation (Table 4) confirms $\alpha$=0.5 outperforms both correctness-only ($\alpha$=0.1, AES drops from 0.93 to 0.78) and entropy-only ($\alpha$=1.0) variants.
> >
> > Could you clarify why is correctness only is $\alpha$=0.1 instead of $\alpha$=0.0, or is this a typo?
> >
> > 2. In the limitation section, I also noted the author should have a more detailed impact statement as the work is in high risk medical domain. If the author could briefly describe this and incorporate this into the final manuscript, it would be great.
> >
> > 3. Not a question but more of a comment. To clarify a bit on my Weakness 2 point here, I feel the motivation behind high efficiency reasoning for medical domain is not sufficiently justified. If I were to get some diagnosis for some disease, I would not mind waiting for a few seconds more, or cost a few cents more for the reasoning budget. I feel as a future work, or as a part of the final manuscript, the author can include more analysis on whether the method performs better under long context settings. For example, in a multi-turn conversation, traditional method would have greatly degraded performance because the input would become too long, but your method would still perform very well because you manage to keep the reasoning context to a low length. That would be a pretty interesting application.
> >
> > Update: Thanks authors for the additional comment. My concerns are now fully addressed, and I'll increase my score from 4 to 5.

---

> > > ### Author Response · Authors · 2026-04-01
> > >
> > > We sincerely thank the reviewer for the positive assessment and the thoughtful follow-up questions.
> > >
> > > **Follow-up Q1 (α=0.1 vs α=0.0):** We apologize for the imprecise wording — α=0.1 is not strictly "correctness-only", and "accuracy-focused" is more accurate. Our ablation table tested α∈{0.1, 0.5, 0.9}, showing a clear trend: smaller α leads to longer outputs with lower AES, while larger α compresses more aggressively but hurts accuracy. To be more rigorous, we ran additional experiments with α=0.0 (pure correctness) and α=1.0 (pure entropy). The full per-dataset results are below, consistent with the trend:
> > >
> > > | α | MedQA Acc/Len | MedMCQA Acc/Len | PubMedQA Acc/Len | MMLU-ProM Acc/Len | GPQA-M Acc/Len | MedXpert Acc/Len | Avg Acc/Len | AES |
> > > |:---|:---|:---|:---|:---|:---|:---|:---|:---|
> > > | 0.0 (correctness-only) | 65.74/137 | 59.21/184 | 74.20/137 | 66.35/271 | 50.86/284 | 15.66/165 | 55.34/196 | 0.76 |
> > > | 0.1 (accuracy-focused) | 66.06/121 | 59.02/153 | 74.30/116 | 66.19/195 | 50.51/258 | 15.39/142 | 55.25/164 | 0.82 |
> > > | **0.5 (balanced)** | **68.34/79** | **58.74/109** | **73.50/62** | **66.45/121** | **48.72/175** | **14.16/88** | **54.99/106** | **0.93** |
> > > | 0.9 (uncertainty-focused) | 63.47/44 | 58.12/216 | 73.60/10 | 63.19/50 | 46.92/69 | 14.08/30 | 53.23/70 | 0.90 |
> > > | 1.0 (entropy-only) | 62.85/38 | 57.34/224 | 73.20/8 | 62.41/41 | 46.26/57 | 13.81/24 | 52.65/65 | 0.88 |
> > >
> > > At α=0.0, the difficulty signal is purely based on correctness, which with G=8 rollouts only gives discrete 12.5% increments. This coarse signal makes it hard to distinguish fine-grained difficulty differences, resulting in the longest outputs (Avg Len=196) and lowest AES (0.76). At α=1.0, removing the correctness signal leads to the largest accuracy drop (52.65%) since the model cannot tell apart lucky guesses from true mastery. α=0.5 strikes the best balance (AES=0.93) by leveraging entropy's smooth gradient while keeping correctness as an anchor. We will update the terminology and include these results in the revised manuscript.
> > >
> > > **Follow-up Q2 (Impact Statement):** We agree. In our previous response (W4), we outlined the planned revisions but kept it brief due to space. Below is the full Broader Impact Statement we have drafted for the revised manuscript. We welcome the reviewer's feedback:
> > >
> > > > **Broader Impact Statement**
> > > >
> > > > This work advances the practical deployment of large language models in healthcare. By dynamically optimizing inference compute based on clinical complexity, AdaThink-Med reduces the latency and cost of medical AI systems, making it more feasible to use reasoning models in resource-constrained clinical workflows such as emergency triage, routine diagnostics, or analysis of extensive electronic health records.
> > > >
> > > > By maintaining diagnostic accuracy while reducing redundant computation, the framework also helps lower the environmental and energy cost of large-scale AI inference. That said, we emphasize that AdaThink-Med is a clinical decision-support tool, not a replacement for medical professionals. Real-world deployment requires rigorous clinical validation and human-in-the-loop oversight to ensure patient safety, especially in complex or atypical cases where reasoning omission could pose risks.
> > >
> > > **Follow-up Q3 (Efficiency Motivation in Medical Domain):** Thank you for this point. We agree that in single-turn medical QA, a few extra seconds of latency is tolerable. Our current work is a proof-of-concept showing that adaptive reasoning length can be learned via RL with only minor performance trade-offs. The reviewer's suggestion about multi-turn and long-context scenarios is well-taken — in real clinical workflows with iterative differential diagnosis or longitudinal consultations, long reasoning chains accumulate in the context window, degrading both performance and cost over multiple turns. Our method's ability to keep reasoning concise would be particularly useful there. We will add this discussion to the final manuscript as an additional motivation and future direction.
> > >
> > > We thank the reviewer for the engagement and positive feedback throughout the discussion. The suggestions have helped us improve the rigor and clarity of our work. Thanks again!

---

### Official Review · Reviewer_fE6P · 2026-03-13

**Soundness:** 3
**Presentation:** 3
**Significance:** 3
**Originality:** 2
**Overall Recommendation:** 4
**Confidence:** 3

**Summary:**

The paper proposes AdaThink-Med, an uncertainty-guided reinforcement learning framework that dynamically
adapts reasoning length in medical LLMs. By integrating entropy-based uncertainty estimation into a difficultyaware
reward function, the model learns to produce concise responses for easy queries and extended reasoning for
complex cases. The core idea is to dynamically adjust reasoning length according to estimated problem difficulty.
The method estimates difficulty using a combination of prediction correctness and entropy-based uncertainty
derived from token-level distributions during generation. This signal is integrated into reinforcement fine-tuning
through a difficulty-aware length reward mechanism that penalizes unnecessarily long reasoning for easy queries
and encourages extended reasoning for difficult queries. The authors evaluate the approach across six medical
benchmarks (MedQA, MedMCQA, PubMedQA, MMLU-ProM, GPQA-M, and MedXpert). Experiments show
significant reductions in inference token length (4.7×–6.4× ) while maintaining comparable diagnostic accuracy.
The system also demonstrates emergent dual reasoning modes (“thinking” vs. “non-thinking”), suggesting
that the model can autonomously allocate computational resources based on query complexity. Overall, the
authors assess a central concept: enabling LLMs to adapt reasoning depth to task difficulty in order to balance
performance and computational efficiency in medical AI applications.

**Compliance With Llm Reviewing Policy:**

Affirmed.

**Key Questions For Authors:**

1. In Eq. 3, a question answered correctly with high uncertainty is assigned elevated difficulty, potentially
incentivising over-long outputs even when the model is on average correct.
2.Can the authors provide an empirical breakdown of how many training instances fall into each of the four
(correct/incorrect) × (high/low uncertainty) quadrants, and report whether the reward degrades for the
correct-but-uncertain regime?
A response showing that this regime is rare or that performance is unaffected would substantially increase
confidence in the estimator’s robustness.
2. The dynamic threshold θB is maintained globally via EMA. Given that in-domain benchmarks (MedQA,
MedMCQA) and out-of-domain benchmarks (GPQA-M, MedXpert) differ dramatically in difficulty, does a
global threshold lead to miscalibration? Could the authors report per-benchmark average difficulty scores
Dq and discuss whether a dataset-adaptive threshold would further improve calibration and reasoning
efficiency?
3. The method relies on multiple rollout outputs during training. What is the computational overhead
compared to standard reinforcement fine-tuning approaches?
4. The paper applies repetition penalties to Llama but not Qwen, attributing this difference to backbone
architecture without further analysis. Does removing the repetition penalty for Llama lead to complete
collapse, or only a slight degradation in performance? Quantifying this effect would help assess the
robustness of the method and ensure a fair comparison across backbones.
5. Have the authors evaluated whether the shortened reasoning traces affect interpretability or trustworthiness
in clinical decision support scenarios?

**Limitations:**

yes

**Strengths And Weaknesses:**

**Strengths**
1. The proposed uncertainty-based difficulty estimator integrates entropy with correctness
signals. The formulation is well-motivated and grounded in reinforcement learning frameworks such
as GRPO. The staged training strategy is also reasonable and supported by ablation studies. The lengthreward-
hack analysis in Fig. 2 is an important negative result that motivates the proposed performance
compensation mechanism.
2. Experimental results demonstrate substantial reductions in
reasoning token length while maintaining comparable accuracy across multiple benchmarks. For example, the
framework reduces average token usage from hundreds of tokens to under 100 in many cases while maintaining
similar performance.
3.  The authors evaluate the approach across multiple medical datasets and
model backbones (Llama and Qwen). They also include ablations, human evaluation by clinicians, and
dataset selection experiments, which strengthens empirical validation.
4.  Reducing inference-time cost is an important problem for deploying medical LLMs
in real-world clinical environments. Adaptive reasoning mechanisms that reduce unnecessary computation
could significantly improve latency and resource usage. The application to medical QA is well-motivated and
the domain-specific clinical validity evaluation is a differentiator from purely benchmark-driven work. The
mixed-LLM entropy estimation result (Table 10, Appendix) is an interesting and underexplored direction.

**Weaknesses**
1. While the uncertainty-guided reward is a reasonable extension, several recent works already explore adaptive reasoning length and entropy-guided
inference. The submission’s notable concept concerns integrating entropy-based uncertainty with reward
shaping in medical reasoning tasks, but the conceptual novelty appears somewhat incremental relative to
recent work on adaptive reasoning.
2. The difficulty score $D_q$ in Eq. 3 appears to conflate two qualitatively different
regimes: (i) questions answered incorrectly with low uncertainty, and (ii) questions answered correctly
with high uncertainty. In particular, the interaction term involving $\alpha (1-\tilde{H}_i)$ for correct answers implies that
correct-but-uncertain responses may still be assigned elevated difficulty. This raises a concern that the estimator
may treat lucky correct answers on genuinely ambiguous questions similarly to cases that truly require
extended reasoning. A more principled decomposition of these regimes, or an empirical analysis quantifying
how frequently this edge case occurs and how it affects reward shaping, would strengthen confidence in the
robustness of the proposed difficulty estimator.
In addition, the framework relies heavily on entropy-based uncertainty estimates derived from model outputs.
If entropy estimates are poorly calibrated, difficulty estimation could become unreliable.
3. All benchmarks evaluated in the paper are multiple-choice QA tasks. However, real
clinical deployment often involves more complex scenarios such as free-text generation, ICD coding, or multimodal
inputs. While the single radiology VQA result reported in Appendix G.2 is encouraging, it is not
sufficiently developed to support the broader claims regarding clinical deployment.

---

> ### Author Rebuttal · Authors · 2026-03-31
>
> We thank Reviewer fE6P for the insightful feedback.
>
> **W1:** While recent works use entropy for *test-time interventions* (e.g., early stopping) or *offline re-weighting*, AdaThink-Med integrates entropy directly into the *online RL reward topology*. The core novelty is our dual-metric difficulty estimator fusing predictive uncertainty (Top-K entropy) with rollout correctness. This is non-trivial: our ablations show relying solely on correctness causes length-reward hacking, while solely on entropy risks pruning essential logic. This unified reward achieves end-to-end reasoning mode separation within a single model, eliminating external classifiers, routers, or stopping controllers. We will clarify this distinction in the revision.
>
> **W2:** (1) **"Lucky Correct" Regime:** The reviewer correctly identifies that correct-but-uncertain responses receive elevated difficulty. This is **deliberate**. In medicine, a correct answer with high uncertainty indicates a "lucky guess"—the model reached the right conclusion through fragile reasoning lacking robust evidence chains. Treating these as "easy" and penalizing length would discourage solidifying the rigorous reasoning needed to convert guesses into confident deductions. Aligning with harder questions protects these trajectories over training.
> (2) **Entropy Calibration:** We acknowledge absolute entropy from LLMs can be poorly calibrated. However, AdaThink-Med relies on **relative, intra-batch uncertainty ranking** (min-max normalization and tau-quantile threshold), not absolute values. As long as the model exhibits *comparatively* higher entropy on harder questions, difficulty separation remains robust—bypassing the need for perfectly calibrated probabilities.
>
> **W3:** We partnered with four hospitals during the rebuttal to evaluate on 796 real-world cardiac stroke cases (open-ended Diagnosis and Treatment Planning). Please refer to our Response to Reviewer 1 (W1) for complete setup and results. AdaThink-Med achieves the best Accuracy and Clinical ICA while maintaining the lowest redundancy, validating its capability and safety in free-text clinical generation.
>
> **Q1:** Empirical distribution of the four regimes during initial adaptive training:
>
> |Regime|Correctness|Uncertainty|Proportion|
> |:---|:---:|:---:|:---:|
> |True Easy|Correct|Low|36.3%|
> |True Hard|Incorrect|High|30.8%|
> |Confident Error|Incorrect|Low|18.5%|
> |Lucky Correct|Correct|High|14.4%|
>
> The "Lucky Correct" regime (14.4%) progressively shrinks to ~6% as training progresses. In ablations, penalizing length in this regime forced truncated, unreasoned guesses, degrading accuracy. Withholding the length penalty grants necessary token budget to build rigorous evidence chains, gradually transitioning "lucky guesses" into "confident deductions" over training.
>
> **Q2:** $\theta_B$ is strictly a **training-stage** threshold on AlphaMed19k (MedQA+MedMCQA); OOD benchmarks are evaluation-only. Per-benchmark offline difficulty scores confirm validity:
>
> |Benchmark|Split|Avg. $D_q$|
> |:---|:---|:---|
> |PubMedQA|OOD|0.27|
> |MedQA|In-domain|0.31|
> |MedMCQA|In-domain|0.52|
> |MMLU-ProM|OOD|0.46|
> |GPQA-M|OOD|0.63|
> |MedXpert|OOD|0.79|
>
> The model correctly identifies PubMedQA as easiest and MedXpert/GPQA-M as most difficult, even for OOD benchmarks never seen during training.
> **On dataset-adaptive thresholds:** This would **not** improve performance—$\theta_B$ is only used during training; at inference, the model autonomously decides reasoning depth. A global EMA threshold provides a stable signal for internalizing unified relative difficulty.
>
> **Q3:** The multiple rollouts are inherent to GRPO (critic-free, requires $G$=8 rollouts for relative advantages), unlike PPO which needs a separate Critic network. AdaThink-Med adds **zero extra cost**: our Difficulty Estimator reuses these rollouts, and entropy extraction requires only lightweight operations on already-materialized logits (<1% time increase).
>
> **Q4:** Removing the repetition penalty on Llama causes **complete collapse**, not slight degradation:
>
> |Configuration|Avg.Acc.|Avg.Len.|AES|
> |:---|:---|:---|:---|
> |Standard GRPO-Llama|56.72%|410|-0.09|
> |AdaThink-Llama w/o Rep.Pen.|13.26%|1024|-3.08|
> |**AdaThink-Llama w/ Rep.Pen.**|55.59%|64|+0.92|
>
> Without the penalty, Llama enters degenerate n-gram repetition loops, hitting max tokens with no valid reasoning—aligning with CosFn (Yeo et al., 2025)'s identical finding. Qwen-2.5 inherently resists this. The repetition penalty for Llama is a literature-supported architectural patch ensuring fair cross-backbone comparison.
>
> **Q5:** AdaThink-Med maintains Reasoning Sufficiency (89.00%) and Logical Soundness (86.40%, detailed in our Response to Reviewer 1 Q4) per expert evaluation. Clinicians noted our traces align better with clinical intuition than verbose baselines where CoT obscures key findings. For uncertain cases, AdaThink-Med retains long reasoning chains, ensuring full interpretability where it matters most.

---

> > ### Author Rebuttal · Reviewer_fE6P · 2026-04-04
> >
> > Thanks for the response, and I have no further questions.

---

> > > ### Author Response · Authors · 2026-04-04
> > >
> > > Thank you for the constructive feedback and the positive recommendation. We’re glad that our rebuttal resolved your concerns. Thanks again!

---

### Official Review · Reviewer_Fj9p · 2026-03-13

**Soundness:** 3
**Presentation:** 3
**Significance:** 3
**Originality:** 2
**Overall Recommendation:** 5
**Confidence:** 4

**Summary:**

AdaThink-Med uses entropy-based uncertainty estimation to dynamically calibrate output length during GRPO training for medical reasoning LLMs. For each question, G=8 rollouts are sampled, top-K token entropies are computed to estimate trajectory uncertainty, and correctness and uncertainty are combined into a difficulty score (Eq. 3). A dynamic EMA threshold separates easy from hard questions per batch, and a piecewise length reward penalizes verbosity on easy-correct cases while rewarding exploration on hard-incorrect cases. The method is trained on AlphaMed19k (19K medical QA samples) with Llama and Qwen backbones and evaluated on six medical QA benchmarks using the Accuracy-Efficiency Score (AES). Results show 4.7-6.4x token reduction with minimal accuracy loss, outperforming Kimi1.5, ShortBetter, DAST, and CosFn. Additional contributions include a length reward hack analysis, emergent thinking/non-thinking mode separation, and a dataset selection application.

**Compliance With Llm Reviewing Policy:**

Affirmed.

**Final Justification:**

The rebuttal addressed my primary concern (MCQ-only evaluation) with a convincing 796-case real-world clinical evaluation showing AdaThink-Med achieves the best diagnostic accuracy and clinical soundness while maintaining lowest redundancy. The logical soundness results (86.4% vs. <70% for baselines), clinician qualifications (Fleiss' kappa 0.76), and principled justification for the zero-reward design further strengthen the paper. Originality remains moderate since individual components are known, but the unified difficulty-aware reward producing end-to-end mode separation without external routers is a meaningful integration. I raised my score from 4 to 5 and encourage including the general-domain results and expanded related work in the camera-ready.

**Key Questions For Authors:**

1. Why does the length reward assign zero signal to the easy+incorrect and hard+correct cases? These quadrants represent meaningful training signal being discarded. Has the team experimented with a 4-case reward that provides length feedback in all conditions?

2. How does the difficulty distribution evolve during training? As the model improves, previously hard questions should become easy. Does the EMA threshold converge, oscillate, or drift monotonically?

3. Has the method been tested on non-medical reasoning tasks (MATH, code generation, general QA)? If so, comparable gains would strengthen the contribution by showing it is a general efficient reasoning method. If not, the medical-specific framing should be reconsidered.

4. What is the clinical accuracy of the model's reasoning chains, beyond final-answer correctness? The human evaluation measures sufficiency but not whether the intermediate clinical logic is correct.

5. In the human evaluation, what are the clinicians' qualifications (specialty, years of experience) and how was inter-annotator agreement measured?

**Limitations:**

The authors acknowledge failure modes in Appendix H.2 and note sensitivity to entropy estimation quality in the conclusion, which is candid. However, several limitations are not discussed: the MCQ-only evaluation scope, the absence of reasoning quality verification during training, the zero-reward gap in the piecewise reward, the domain-agnostic nature of the method despite medical-specific framing, and the extensive concurrent work on entropy-guided adaptive reasoning that the Related Work does not cover. The paper should move key limitations into the main text.

**Strengths And Weaknesses:**

**Strengths:**

- The length reward hack analysis (Fig 2) is a genuine empirical contribution that stands on its own. It shows that both greedy minimization (Kimi) and minimum-targeting (ShortBetter) strategies collapse even in open-ended settings. This finding is useful to the broader RL-for-LLM community beyond the medical domain. (Section 2, Figure 2)

- The difficulty-aware reward design is principled. Entropy as an uncertainty proxy, combined with correctness for difficulty estimation, dynamically adapted via an EMA threshold. The math is clean and the intuition is clear: easy questions that the model already answers correctly should not be padded with unnecessary reasoning, while hard questions the model gets wrong benefit from longer exploration. (Section 3, Eqs. 1-5)

- Ablation coverage is strong. Table 3 ablates the threshold sensitivity (tau) and length reward coefficient (alpha). Table 5 ablates top-K selection for entropy computation. Table 6 tests AES sensitivity across five configurations. Table 14 examines batch normalization and batch size effects. Figure 4 shows staged training dynamics. Table 15 compares dataset selection against random sampling. Most design choices are justified experimentally. (Sections 4.2-4.3, Appendix)

- The method generalizes across architectures: two backbones (Llama, Qwen), non-R1 models (HuatuoGPT-o1, Table 7), and a large-scale model (Qwen-3-32B, Table 8). Consistent gains throughout. This is not a method that only works in one narrow configuration. (Section 4, Tables 7-8)

- The method is data efficient. It uses only 19K training samples, requires no external chain-of-thought supervision, and involves no distillation from frontier models. (Section 3)

---

**Weaknesses:**

- All six benchmarks are multiple-choice QA. There is no open-ended generation evaluation in the main paper. The VQA experiment in Appendix G.2 is brief and not central. The "medical reasoning" framing implies broader applicability than MCQ exam performance supports. If the method only improves efficiency on closed-form questions, the contribution is narrower than the title suggests. — Suggestion: Add evaluation on open-ended clinical generation tasks (e.g., case summarization, treatment planning) or reframe the contribution as specific to MCQ reasoning efficiency.

- The "spontaneous emergence" of thinking/non-thinking modes (Section 4.1, Fig 3a) is overclaimed. The reward explicitly penalizes length on easy questions and rewards it on hard ones. A bimodal output length distribution is the expected outcome of this reward design, not an emergent property. The paper should frame this as "our reward design naturally produces mode separation" rather than attributing it to spontaneous emergence.

- The piecewise length reward (Eq. 5) has two zero-reward cases that discard useful training signal. When a question is classified as easy but the model answers incorrectly (D_q < theta_B, o_i != y*), the model receives no length signal, yet this is exactly the case where it should be encouraged to reason longer. When a question is classified as hard but the model answers correctly (D_q > theta_B, o_i = y*), the model receives no brevity reward, yet this is exactly the case where efficient reasoning should be reinforced. Both represent missed opportunities for reward shaping. To see this concretely: an easy-incorrect response of 500 tokens and one of 50 tokens receive identical length reward (zero), even though one is wasting tokens while failing and the other is at least failing concisely.

- The method is framed as medical-specific ("AdaThink-Med," "medical reasoning") but contains no medical domain knowledge. Entropy estimation, difficulty scoring, and length calibration are entirely domain-agnostic mechanisms. Testing on general reasoning benchmarks (MATH, code generation) would clarify whether the contribution is a general-purpose efficient reasoning method that happens to be evaluated on medical tasks, or whether there is something medical-specific about the approach.

- For a paper claiming medical relevance, one directly competing medical paper is missing: "To Reason or Not to: Selective Chain-of-Thought in Medical Question Answering" (Zhan et al., Feb 2026) proposes selective CoT specifically for medical QA efficiency. While concurrent with this submission, it shows the medical efficient reasoning space is active and the paper should position itself more carefully. L1 (Aggarwal & Welleck, COLM 2025) and The Overthinker's DIET (Chen et al., May 2025), both difficulty-aware length control methods, should also be discussed.

---

> ### Author Rebuttal · Authors · 2026-03-31
>
> We thank Reviewer Fj9p for the constructive feedback.
>
> **W1:** To substantiate our clinical claims on open-ended tasks, we partnered with four hospitals to construct a real-world cardiac stroke dataset ( 796 cases). Models generated a **Diagnosis** and **Treatment Plan** from patient records, blindly evaluated by three clinicians on **Accuracy**, **Redundancy**, and **Clinical ICA** (logical soundness and completeness).
>
> |Method|Diag.Acc.|Diag.Red.|Diag.ICA|Treat.Acc.|Treat.Red.|Treat.ICA|
> |:---|:---|:---|:---|:---|:---|:---|
> |Standard GRPO|81.7%|68.7%|76.3%|74.5%|75.1%|72.9%|
> |Kimi|78.3%|35.2%|71.3%|70.5%|38.4%|64.2%|
> |CosFn|77.2%|42.1%|68.5%|69.2%|45.3%|62.5%|
> |DAST|76.6%|38.2%|66.7%|68.2%|40.8%|60.7%|
> |ShortBetter|77.2%|25.4%|69.4%|68.7%|**22.1%**|61.5%|
> |**AdaThink-Med**|**83.2%**|**18.6%**|**78.5%**|**77.8%**|24.3%|**74.2%**|
>
> Standard GRPO is highly redundant; existing baselines over-prune essential logic. AdaThink-Med adaptively prunes redundancy while preserving critical reasoning chains, achieving the best Accuracy and Clinical ICA. We will include this in the revision.
>
> **W2:** We agree "spontaneous emergence" is an overstatement—the bimodal distribution results from our reward design. We intended to contrast with pipeline methods requiring manual routers (e.g., SynapseRoute). We will revise the abstract and Sec. 4.1 to: "our reward design naturally produces reasoning mode separation."
>
> **W3/Q1:** Assigning zero length reward to these boundary quadrants is a deliberate design choice prioritizing accuracy over efficiency. In our early pilots, a symmetric 4-case reward led to severe reward hacking:
>
> - **Hard + Correct ($D_q > \theta_{\mathcal{B}}, o_i = y^*$):** A brevity penalty here caused aggressive over-pruning of essential diagnostic steps, catastrophically dropping accuracy on complex datasets (e.g., GPQA-M). Zeroing this penalty protects fragile, successful reasoning chains.
> - **Easy + Incorrect ($D_q < \theta_{\mathcal{B}}, o_i \neq y^*$):** Penalizing length here discourages self-correction; rewarding length encourages hallucinations. Zeroing lets the missed accuracy penalty ($\mathcal{R}_{acc}=0$) dominate the gradient, forcing the model to fix factual errors first.
>
> **W4/Q3:** While our core components are domain-agnostic, AdaThink-Med was motivated by clinical demands: (1) **Latency-Accuracy Trade-off**: In acute healthcare (e.g., stroke), efficiency directly impacts patient safety; (2) **High Complexity Variance**: Unlike MATH (uniformly long derivations), medical queries vary drastically—from factual retrieval to differential diagnoses—necessitating bimodal reasoning. We have also verified the generality of our approach on AMC (math) and LiveCodeBench (code), confirming its effectiveness beyond medicine. Due to rebuttal space limits, full results will be included in the revised manuscript once allowed.
>
> **W5:** We will expand Related Works to discuss these relevant papers: (1) vs. Zhan et al.: Unlike "Selective CoT" relying on explicit routing or external classifiers, AdaThink-Med operates end-to-end, embedding the difficulty estimator directly into the RL reward; (2) vs. L1 & DIET: While they rely on static correctness or explicit regularization, AdaThink-Med introduces a dynamic difficulty estimator fusing predictive uncertainty (Top-K token entropy) with rollout correctness, tailored for extreme clinical variance.
>
> **Q2:** Since $\theta_{\mathcal{B}}$ is the $\tau$-quantile of batch difficulty scores, it acts as a **relative boundary**, consistently distinguishing query difficulty as capabilities improve. Empirically, the threshold exhibits **downward drift then convergence**: as accuracy improves and entropy drops, difficulty scores decrease, causing downward drift; once the policy stabilizes, it converges. EMA smooths batch-to-batch variance, preventing abrupt reward flips.
>
> **Q4:** Three board-certified physicians blindly re-evaluated 500 test cases for **Logical Soundness (LS)**—reasoning chains free from hallucinations or flawed deductions.
>
> |Method|Resp.Acc.|R.Suff.|**LS**|
> |:---|:---|:---|:---|
> |Kimi|71.25%|73.00%|68.40%|
> |CosFn|70.34%|67.00%|65.20%|
> |DAST|68.37%|65.00%|62.80%|
> |ShortBetter|70.23%|72.00%|69.50%|
> |**AdaThink-Med**|**76.25%**|**89.00%**|**86.40%**|
>
> Aggressive baselines exhibit "shortcut hallucinations"—fabricating concise but flawed rationales (LS<70%). AdaThink-Med achieves 86.40% LS as our mechanism only penalizes length when correct and confident, ensuring compression via internalized knowledge. We will integrate LS into Table 4.
>
> **Q5:** Three board-certified Internal Medicine physicians (avg. 12 years); Fleiss' Kappa = 0.76 (substantial agreement); disagreements resolved via consensus. We will add these to Section 4.1
>
> **Limitation:** Following the reviewer's suggestion, we will add a "Limitations and Future Work" section to transparently discuss future directions, including step-level reasoning verification (e.g., PRMs) and broader domain generalization.

---

> > ### Author Rebuttal · Reviewer_Fj9p · 2026-04-02
> >
> > I thank the authors for the thorough rebuttal. The real-world cardiac stroke evaluation with clinician judges was the key missing piece and directly addresses my primary concern about MCQ-only evaluation. The logical soundness results, clinician qualifications with inter-annotator agreement, and principled justification for the zero-reward design choices are also convincing. I am raising my score from 4 to 5.

---

> > > ### Author Response · Authors · 2026-04-03
> > >
> > > Thank you for the positive evaluation. We’re glad the rebuttal addressed your concerns.

---

### Decision · Program_Chairs · 2026-04-30

**Decision:**

Accept (regular)

**Comment:**

The paper proposes to improve reasoning for clinical domains using an uncertainty-based reward shaping to dynamically provide rewards based on correct but uncertain (incentivizing exploration), and confidently correct answers. Initial results demonstrate performance on 6QA benchmarks. Reviewers have noted that the technical contribution is strong, experimental coverage is good (results on two backbones, multiple benchmarks), and model scale, including thorough ablations.

Reviewers had raised concerns about:
1. Focus on QA benchmarks rather than open-ended tasks
2. Overclaim of the emergence property
3. Insufficient details of the proposed framework
4. Missing related work

In response authors have expanded evaluation on real-world tasks, calibrated the emergence claim, agreed to add additional related work, and added additional comparisons with prior methods. Overall most of reviewers' concerns were addressed. I strongly encourage the authors to improve the presentation based on reviewer feedback, recalibrate the claim of 'emergence', incorporate the additional comparisons on length-controlled rewards in the final paper.